# Rethinking Label Poisoning for GNNs: Pitfalls and Attacks

**Vijay Lingam**[1]       **Mohammad Sadegh Akhondzadeh**[2]       **Aleksandar Bojchevski**[2]

[1] CISPA Helmholtz Center for Information Security       [2] University of Cologne

`vijaylingam0810@gmail.com, {makhondz,a.bojchevski}@uni-koeln.de`

## Abstract

Node labels for graphs are usually generated using an automated process or crowd-sourced from human users. This opens up avenues for malicious users to compromise the training labels, making it unwise to blindly rely on them. While robustness against noisy labels is an active area of research, there are only a handful of papers in the literature that address this for graph-based data. Even more so, the effects of adversarial label perturbations is sparsely studied. More critically, we reveal that the literature on label poisoning for GNNs is plagued by evaluation pitfalls. Thus making it hard to conclude how robust GNNs are against label perturbations. After course correcting the state of label poisoning attacks with our faithful evaluation, we identify a discrepancy in attack efficiency of $\sim 9\%$ on average. Additionally, we introduce two new simple yet effective attacks that are significantly stronger (up to $\sim 8\%$) than the previous strongest attack. Our strongest proposed attack can be efficiently computed[1] and is theoretically backed.

## 1 Introduction

Graph Neural Networks (GNNs) have emerged as a powerful tool to learn from graph-structured data. From social network analysis and recommendation systems, to bio-informatics and traffic prediction, their wide-ranging applications highlight their importance (Wu et al., 2019b). Consequently, we need to understand their robustness, especially if we aim to integrate them in safety-critical domains. There is a large body of work showing that GNNs are susceptible to feature and structure perturbations, which can significantly degrade their performance (see e.g. surveys by Sun et al. (2023), Chen et al. (2020), and Jin et al. (2021)). This vulnerability exposes them to potential adversarial attacks, thereby compromising the integrity of their outputs. Even if there is no adversary, we argue that understanding the worst-case robustness of a model can be highly informative, especially when comparing models.

Among the various attack vectors, label poisoning poses a distinct threat. In many real-world scenarios, labels for training data are often generated through crowd-sourcing or other non-expert sources. For example, in a federated learning setting where multiple peers collaboratively train a shared model, any peer may submit potentially poisoned labels. Similarly, models are often trained on data scraped from the internet without careful quality control. This opens up an avenue for attackers to implant poisoned labels, thereby manipulating the learning process of GNNs. Label poisoning attacks can be subtle and challenging to detect, thereby complicating defenders' efforts to mitigate them. This is especially true for graphs since we cannot easily "see" whether the node's label is correct, unlike for e.g. images. Such attacks can be specifically crafted to manipulate recommendation or advertising systems, granting a competitive advantage (Fang et al., 2018; Zhang et al., 2021).

Recent work by Carlini et al. (2023) demonstrates the practicality of data poisoning attacks for crowd-sourced and publicly-crawled datasets. The strength of the poisoning attack depends on the data available to the attacker – access to full/partial training labels/graph structure. For illustration purposes, we compute the optimal label flipping attack for the GCN model (Kipf & Welling, 2017) on a tiny version of the Cora-ML dataset by exhaustively enumerating all possible label flips (more details in the § B.1). We observe that only a single adversarial label flip can drop GCN's test performance from $81.27_{\pm 5.64}$ to $64.36_{\pm 6.26}$, a staggering drop of $\sim\textbf{17\%}$.

---

[1] `https://github.com/VijayLingam95/RethinkingLabelPoisoningForGNNs`

We will show that similar results hold in general – across all datasets and models, a relatively few flips are enough to significantly affect performance. Despite its potential impact, label poisoning for GNNs has been relatively under-explored in the research community, leaving a critical gap in our understanding of their robustness. While label poisoning is better understood for non-graph data (Biggio et al., 2011; Jin et al., 2021), graphs come with unique challenges due to label sparsity and the interdependence between nodes. We aim to fill this gap by conducting a comprehensive robustness analysis of GNNs under label poisoning, making *four* key contributions.

1. We pinpoint significant *pitfalls* in the evaluation setup of previous studies on label poisoning. These pitfalls make it nearly impossible to accurately interpret the results of previous studies, resulting in a distorted portrayal. The root issue is that prior assessments fail to accurately replicate the actions of a defender. By rectifying these pitfalls, we establish a more reliable and faithful evaluation framework for gauging the robustness of GNNs against label poisoning.

2. We propose two simple, yet highly effective, families of baseline attacks that stem from different approximations of the (NP-hard) bi-level optimization problem associated with label poisoning. Our two strategies, each with different advantages, rely on *linear surrogates* and *meta learning* respectively. These attacks are significantly stronger than previous ones, affecting both vanilla and noise-aware models. This demonstrates the pressing demand for more robust defense mechanisms, especially considering the *transferability* of our attacks, wherein a strategy devised for one model can effectively contaminate numerous other models.

3. We shed light on a curious phenomenon where unconstrained attacks perform worse than binary attacks, where the attacker is constrained to only flip among two classes. In addition to experimental evidence, we show that random binary flips lead to provably lower accuracy.

4. We show that the mixed integer linear program (MILP) associated with our linear surrogate attack can be efficiently solved by proving that its LP relaxation is integral. Moreover, we provide a closed-form optimal solution for the binary variant which can be computed even faster.

## 2   BACKGROUND: THREAT MODEL, EXISTING ATTACKS & RELATED WORK

We focus on label poisoning attacks for semi-supervised node classification. The adversary can perturb the labels of a small fraction of training nodes, keeping the features and the structure unperturbed.

**Problem setting.** We are given a graph represented by its $|\mathcal{V}| \times |\mathcal{V}|$ adjacency matrix $\boldsymbol{A}$ and its $|\mathcal{V}| \times D$ feature matrix $\boldsymbol{X}$, where $\mathcal{V}$ is the set of nodes. The $D$-dimensional features can be discrete or continuous, and the graph can be directed, undirected, weighted or unweighted. We represent the ground-truth labels as one-hot encoded row vectors in a matrix $\boldsymbol{Y} \in \{0,1\}^{|\mathcal{V}| \times C}$ where $C$ is the number of classes. We denote with $\boldsymbol{Y}_l$ and $\boldsymbol{Y}_u$ the labels corresponding to the subset of training (labeled) nodes $\mathcal{V}_l$ and the test (unlabeled) nodes $\mathcal{V}_u$. Given $\boldsymbol{A}, \boldsymbol{X}, \boldsymbol{Y}_l$ we learn the parameters $\boldsymbol{\theta}^*$ of a model $f(\boldsymbol{\theta})$, e.g. a GNN, by minimizing some loss $\mathcal{L}$ (usually cross-entropy).

**Threat model.** The attacker can perturb the training labels $\boldsymbol{Y}_l$ to reduce the performance on the test nodes. We assume that the attacker has complete knowledge of $\boldsymbol{X}$. We consider three settings where the attacker knows: (i) $\boldsymbol{A}$ and all ground-truth labels $\boldsymbol{Y}$; (ii) $\boldsymbol{A}$, the ground-truth labels for the training nodes $\boldsymbol{Y}_l$ only, and predicted labels for the test nodes $\widehat{\boldsymbol{Y}}_u$; and (iii) no access to $\boldsymbol{A}$ and all ground-truth labels $\boldsymbol{Y}$. We focus on (i) and defer (ii) and (iii) to § B.8. In all settings, the attacker is only allowed to perturb $\boldsymbol{Y}_l$. The first setting corresponds to the worst-case scenario and is important for understanding the intrinsic robustness of our models, regardless of whether it is feasible to execute such an attack in practice. We can write down the label poisoning attack as a bi-level optimization problem:

$$\boldsymbol{H}^* = \arg\max_{\boldsymbol{H}} \quad \mathcal{L}(\boldsymbol{\theta}^*; \boldsymbol{A}, \boldsymbol{X}, \boldsymbol{Y}_u)$$
$$\text{s.t.} \quad \boldsymbol{\theta}^* = \arg\min_{\boldsymbol{\theta}} \mathcal{L}(\boldsymbol{\theta}; \boldsymbol{A}, \boldsymbol{X}, \boldsymbol{H}), \quad \|\boldsymbol{Y}_l - \boldsymbol{H}\|_0 \leq 2\epsilon|\mathcal{V}_l| \tag{1}$$

where $\boldsymbol{H}$ denotes the potentially poisoned training labels, $\epsilon \in (0,1)$ is the perturbation budget, and $\lfloor \epsilon |\mathcal{V}_l| \rfloor$ is the maximum number of labels that can be flipped. The inner problem corresponds to training a model with potentially poisoned training node labels, while the outer problem maximizes the loss on the test nodes, given the optimal parameters $\boldsymbol{\theta}^*$. Optimally solving the bi-level problem in Eq. 1, or often even just the inner problem itself, is intractable (NP-hard) in general. Attacks differ in how they construct an approximation to this problem.

**Defense model.** The defender is given a dataset with potentially poisoned labels, but does not know which labels are poisoned. Since we're dealing with poisoning, the defender trains the model from scratch. This could be a vanilla model or a noise/poison-aware robust model. Depending on the model they can use all available labels for training (e.g. label propagation) or split them into training and validation sets (e.g. GNNs). Often the defender also performs hyper-parameter tuning before deployment since GNNs are sensitive to h-params (Shchur et al., 2018; Li & King, 2020).

**Heuristic-based attacks.** The *random label-flipping attack* (**RND**) randomly selects a subset of training nodes within the budget, and then randomly selects incorrect labels for each selected node. The *degree-based label-flipping attack* (**DEG**) selects the highest degree nodes given a budget, and randomly assigns incorrect labels. As we will show in § 5 even these simple baselines can outperform some prior attacks under a fair evaluation.

**Learning-based attacks.** All previous attacks approximately solve Eq. 1 by using a fixed surrogate model. They all replace the inner problem with a closed-form solution, but differ in how they tackle the outer problem. The poisoned labels are then applied to a target model.

Liu et al. (2019) propose a gradient-based *label propagation attack* (**LP**) for graph-based semi-supervised learning (G-SSL) models. Note that this attack was **primarily** designed for binary-class datasets that are i.i.d in nature, with G-SSL methods applied on top. Nonetheless, it can be adapted to our graph setting. LP replaces the inner optimization in Eq. 1 with the closed-form solution of label propagation. They replace the loss in the outer problem with an expectation over learnable Bernoulli variables that model the probability of flipping the label of a given node. They optimize this new loss using a reparametrization trick and gradient descent.

Zhang et al. (2020) build on top of the LP attack and propose LAFAK (**LFK**) that replaces the inner optimization in Eq. 1 with a regression-based closed-form solution of a linearized GCN. They remove the non-linearities in a 2-layer GCN (Kipf & Welling, 2017): SOFTMAX($\widehat{A}$ RELU($\widehat{A} X \theta_1$) $\theta_2$) → SOFTMAX($\widehat{A}^2 X \theta$), where $\widehat{A}$ is the symmetric normalized adjacency matrix, and $\theta_1$ and $\theta_2$ are reparameterized into a single matrix $\theta \in \mathbb{R}^{D \times C}$. Here, the (implicit) surrogate model is SGC (Wu et al., 2019a). Next, they replace the cross-entropy loss with a least squares surrogate loss, using regression to perform classification. That is, $\theta^* = \arg\min_\theta \|\widehat{X}_l \theta - \hat{y}_l\|_2^2 + \lambda\|\theta\|_2^2$, where $\hat{y}_l \in \{-1, +1\}^{|\mathcal{V}_l|}$ is the vector of (potentially poisoned) *binary* training labels, $\widehat{X} = \widehat{A}^2 X$ are the diffused features, $(\cdot)_l$ select the subset of rows corresponding to training nodes, and $\lambda$ is the regularization strength. Now, $\theta^* = (\widehat{X}_l^T \widehat{X}_l + \lambda I)^{-1} \widehat{X}_l^T \hat{y}_l$. Similar to LP, they substitute the non-differentiable parts of the outer problem with continuous surrogates and use a gradient descent-based optimizer.

Liu et al. (2022) propose the *maximum gradient attack* (**MG**) based on label propagation. Unlike LP which uses a similarity matrix that is constructed by applying a Gaussian kernel to the feature matrix, MG proposes multiple ways to construct a propagation matrix $\bar{A}$ (e.g., PageRank matrix, higher-order adjacency matrix). Then, they greedily select the top nodes within the budget with the highest gradient w.r.t. the outer loss, and set their label to the false class with the highest index.

**Binary vs. multi-class.** The default implementation of LFK and LP support only binary labels. Thus, for multi-class datasets they consider only the subset of the graph whose nodes belong to the two most frequent classes, restricting their attack to flips among these two classes. Note, this restriction is only for the attack phase, during evaluation the entire (multi-class) graph is used. It turns out that this restriction to binary flips may strengthen the attack efficacy, as suggested by our theory in § 4.3. As a side note, this binary approach may limit the maximum perturbation budget. In § B.6 we discuss this issue in detail and propose a minor baseline extension to alleviate it.

**Discussion.** All these attacks have shortcomings. The LP attack was primarily built for G-SSL datasets that do not contain a graph, limiting its efficacy on message-passing based GNNs. LFK relies on several continuous surrogates to enable gradient flow. These approximations hurt its performance. However, coincidentally, restricting to binary flips boosts its attack performance. MG's greedy approach can be myopic. While these existing label attacks claim severe degradation in GNNs performance with minimal label perturbations, we identify serious flaws in their evaluation setup. In the following section (§ 3), we discuss and fix these pitfalls and perform a faithful evaluation of the existing label attacks. We provide further discussion of broader related work in § B.2.

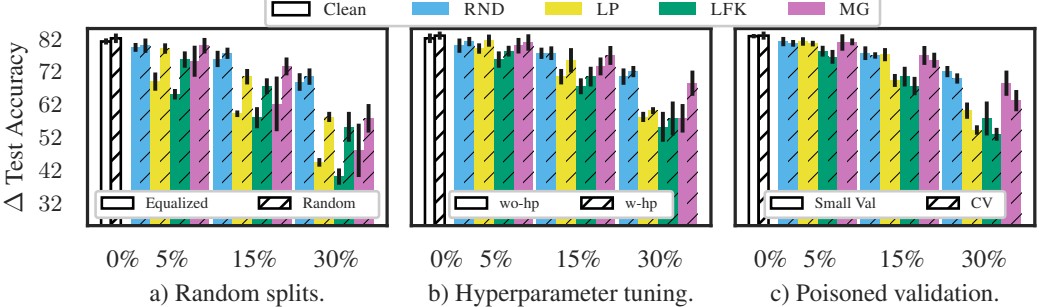

Figure 1: Previous evaluations use class-equalized splits (P4), untuned models (P5), and unpoisoned (clean) and unreasonably large val set (P1, P6). We fix these pitfalls in (a), (b), and (c) respectively.

## 3 SIX PITFALLS AND REMEDIES

After careful inspection, we conclude that the evaluation of previous attacks does not simulate the full potential of a defender. In short, the dataset splits and the training/tuning routine simulated by the attacker when evaluating the strength of their proposed attacks in the previous works we examined is not realistic and flawed. In the rest of this section, we carve out the specifics of the pitfalls we identified, empirically validate them with our experiments, and offer remedies. Here we show results on the Cora-ML (McCallum et al., 2000) dataset for the GCN (Kipf & Welling, 2017) model. In § B we provide detailed information on the setup. In § B.17 we provide further experiments on additional datasets and models which lead to the same conclusions. For the sake of completeness, we sweep the poisoning budget across the range [5%, 10%, 15%, 20%, 30%] of training dataset. For the Cora-ML dataset these budgets correspond to [7, 14, 21, 28, 42] training labels out of 2810 in expectation. In practice, smaller budgets are likely to be more relevant. We plot our findings in Fig. 1.

**First three pitfalls.** We focus on the common *sparsely labeled* scenario where the fraction of train nodes is relatively small, $|\mathcal{V}_l| \ll |\mathcal{V}_u|$. **P1.** The first pitfall arises since all previous attacks evaluate on data splits with validation set size much larger than that of the training set (e.g. 500 validation vs 140 train nodes on Cora-ML in the default setting). Oliver et al. (2018) offer an extensive discussion on this pitfall for general semi-supervised learning (see their P.6). Briefly, in real-world applications the larger val set would instead be used as the train set. Also, h-params tuning would be noisier when using a realistically small validation set. **P2.** The second pitfall is the failure to report the standard deviation across multiple splits. This is crucial since GNNs are highly sensitive to the split, especially for sparsely labeled data. We remedy these pitfalls by using a validation and train set of the same size, and by reporting the mean and standard deviation across 10 splits (more details in § B.8). **P3.** The third pitfall, is to only evaluate attacks on undefended models. We find that noise/poison-aware models can offer some robustness against these attacks. We explore this in detail in § 5.

**P4. Class-equalized vs. random split.** In the (default) class-equalized setting both the train and val sets contain equal number of samples *per class*. This does not preserve the global class distribution and treats each class equally. Another potential way to split the dataset is to randomly sample nodes for train and val sets (referred to as random setting). We argue that this is more realistic, as the train/val class distribution would more closely align with the test/true class distribution. From Fig. 1(a), we deduce that switching to the random setting significantly degrades the performance of label-poisoning attacks. Compared to the learning-based attacks which show a difference of up to 17%, the heuristic attacks (RND and DEG) are less susceptible to changes in data splits. LFK and LP attacks flip pairs of popular classes, and MG attack always flips to the label with the highest index. The class distribution in all studied datasets is skewed with 2 or 3 majority classes (see § B.15). Since the random setting has more samples from popular classes, the model has better chances at learning a more robust decision boundary and thus weakens the attack's efficacy.

**P5. Hyper-parameter tuning.** GNNs are usually highly sensitive to hyper-parameters (h-params). Even though this is well established (Li & King, 2020), all previous poisoning studies still use the default h-params to evaluate the effectiveness of their proposed attack. Since the default h-params were tuned to work with clean (unpoisoned) labels they might be suboptimal. On Fig. 1(b) we show

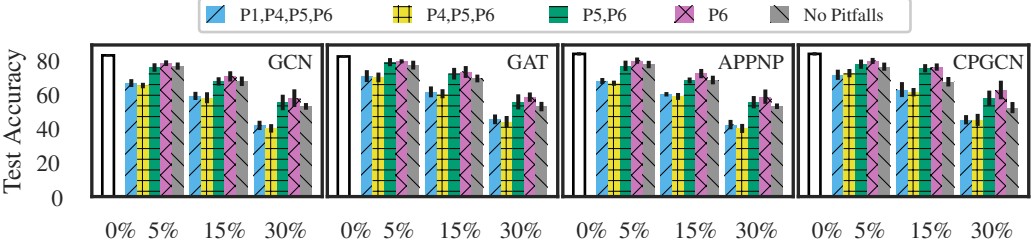

Figure 2: Each bar corresponds to an evaluation setup with a subset of pitfalls (LFK, Cora-ML). Pitfall P2 is fixed since we report standard deviations. P3 is fixed by evaluating the CPGNN defense.

the performance of various attacks with and without tuning. Fine-tuning with only 20 configurations (see § B) can significantly deteriorate the attack, manifested as increase in test accuracy. The recovery in performance is more pronounced in learning-based attacks and at higher poison percentages. At 5% and 15% budget the claimed state-of-the-art attack, MG, is as ineffective as the RND baseline. We observe this anomaly across models and datasets as documented in § B.8. LFK seems to be the strongest attack followed by LP. Since tuning GNNs is relatively cheap (Table 13) and since the untuned model performs poorly, it is a reasonable assumption that the defender will tune their models.

**P6. Clean validation set.** All existing attacks assume in their evaluation that the validation set labels are clean (not poisoned). We hypothesize that having a clean (and large, see first pitfall) validation set can aid the model in recovering accuracy by avoiding overfitting on the poisoned train set. Additionally, it is unrealistic to treat the train and validation sets separately – if the defender has a clean validation set they can simply ignore the poisoned train set (see § B.10 for a detailed analysis). Note, unlike e.g. image classification, for semi-supervised node classification both train and val set are usually small since $|\mathcal{V}_l| \ll |\mathcal{V}_u|$, and often have equal size. To remedy this pitfall, we poison the available sparse set of labels $\mathcal{V}_l$ which the defender can use for training and/or validation as they see fit.[2] Since we train GNNs, we create 10 different train/val sets from the poisoned $\mathcal{V}_l$ and report the average across splits. We refer to this as the CV setting, and on Fig. 1(c) we see that it boosts the efficacy of all attacks. This is expected, since the defender no longer has a clean validation set.

**Cumulative effect of the pitfalls.** In Fig. 2, we sequentially apply the series of remedies and plot LFK's performance. Moving from class-equalized splits to random splits has the largest impact on average, followed by hyper-parameter tuning. Comparing the first and last bar for a given budget shows that the attack performance on average worsens, e.g. for the LFK attack on Cora-ML by ∼9% across models. Additional comparisons are provided in § B.8 which lead to the same conclusions. We advocate for using our proposed evaluation procedure as it better reflects real scenarios. In § 4 we introduce our new attacks, which we compare against these baselines in § 5.

## 4  LABEL POISONING ATTACKS

Motivated by the results in § 3 where we showed that previous attacks are not as effective as claimed, we design new simple and effective baseline attacks. These attacks are independent from the pitfalls but we discuss some connections in § B.18. We propose two different strategies to approximate the bi-level poisoning problem in Eq. 1 that rely on: (i) linear surrogate models and (ii) meta learning.

### 4.1  LINEAR SURROGATE ATTACK

We start by replacing the inner optimization problem with a linear surrogate for which we can obtain the optimal weights $\boldsymbol{\theta}^*$ in closed-form. We explore two different variants. First, similar to Zhang et al. (2020) we use an SGC (Wu et al., 2019a) surrogate which is equivalent to linearizing a vanilla 2-layer GCN (Kipf & Welling, 2017). We use SGC, since despite its simplicity it shows similar performance to its non-linear counterpart across datasets. Second, we use the Neural Tangent Kernel (NTK)

---
[2]Some methods, such as label propagation, use all labels for training, while others like GNNs split $\mathcal{V}_l$ into train/val sets and use the latter for early stopping/fine-tuning. Our attacks remain agnostic to this.

for an infinitely-wide GCN (Sabanayagam et al., 2022). We use the NTK since it is theoretically well-founded and it has been shown to capture the behavior of its finite-width counterpart.

In both cases, unlike Zhang et al. (2020) which only support binary labels, we replace the inner problem with $\boldsymbol{\theta}^* = \arg\min_{\boldsymbol{\theta}} \|\widehat{\boldsymbol{X}}_l \boldsymbol{\theta} - \boldsymbol{H}\|_2^2 + \lambda\|\boldsymbol{\theta}\|_2^2$. In this surrogate *multi-output* ridge regression problem, $\boldsymbol{\theta} \in \mathbb{R}^{D \times C}$, we regress against the one-hot labels $\boldsymbol{H}$, thereby supporting multi-class problems. For the SGC variant $\widehat{\boldsymbol{X}} = \widehat{\boldsymbol{A}}^2 \boldsymbol{X}$, and for the NTK variant $\widehat{\boldsymbol{X}}$ is the kernel matrix derived by Sabanayagam et al. (2022)(see § B.4 for more detail). The closed-form solution is readily obtained as $\boldsymbol{\theta}^* = \widetilde{\boldsymbol{X}}_l \boldsymbol{H}$ where $\widetilde{\boldsymbol{X}} = (\widehat{\boldsymbol{X}}^T \widehat{\boldsymbol{X}} + \lambda \boldsymbol{I})^{-1} \widehat{\boldsymbol{X}}^T$ is a constant that can be pre-computed once[3] for each dataset. Given our insights from pitfall P5 we choose the $\lambda$ that maximises the validation accuracy on average across splits, preventing overfitting to a single split. Now, we rewrite the surrogate variant of Eq. 1 as the following mixed-integer linear program (MILP):

$$\min_{\boldsymbol{H} \in \{0,1\}^{L \times C}} \mathcal{L}(\boldsymbol{Y}_u, \widehat{\boldsymbol{Y}}_u) \tag{2a}$$

$$\|\boldsymbol{H} - \boldsymbol{Y}_l\|_0 \leq 2\epsilon L \tag{2b}$$

$$\widehat{\boldsymbol{Y}}_u = \widehat{\boldsymbol{X}}_u \widetilde{\boldsymbol{X}}_l \boldsymbol{H} \tag{2c}$$

$$\boldsymbol{H} \mathbf{1}_C = \mathbf{1}_L \tag{2d}$$

where $\boldsymbol{H}$ is a binary variable, $L = |\mathcal{V}_l|$ is the number of training nodes, and $\mathbf{1}_L$ is the all-ones vector of size $L$. Here, Eq. 2b enforces the budget constraint, Eq. 2c computes the predictions $\widehat{\boldsymbol{Y}}_u$ for the test nodes, and Eq. 2d enforces a one-hot encoding. We explore several variants for the loss in Eq. 2a. One choice is $\mathcal{L}(\boldsymbol{Y}_u, \widehat{\boldsymbol{Y}}_u) = \text{sum}(\boldsymbol{Y}_u \odot \widehat{\boldsymbol{Y}}_u)$ which is equivalent to minimizing the predicted "probability" for the true class. The benefit of this loss is that it is linear in $\boldsymbol{Y}_u$ and $\widehat{\boldsymbol{Y}}_u$. Alternatively, we can compute the most likely label for each node, by computing the $\arg\max$ over the rows of $\widehat{\boldsymbol{Y}}_u$. This requires additional auxiliary binary variables to encode the $\arg\max$, making it inherently less scalable.

The MILP problem in Eq. 2 finds the optimal subset of nodes to perturb and which labels to perturb them to (via $\boldsymbol{H}$). Next, we show that, surprisingly, the *optimal* poisoned labels can be efficiently computed by calling any LP solver (we use MOSEK). The proof § A exploits total unimodularity.

**Proposition 1.** *The optimal solution of Eq. 2 is integral, i.e. $\boldsymbol{H}^* \in \{0,1\}^{L \times C}$, when relaxing the MILP to an LP such that $\boldsymbol{H} \in [0,1]^{L \times C}$.*

We also explore another variant where we fix the target label, e.g. the false label with the highest probability as predicted by a clean model (see § B.5 for details). We call the first variant SGC and the second SGC-FIX. For SGC-FIX we only have to find the optimal subset of nodes which should be significantly easier. In fact, we have the following closed-form solution.

**Proposition 2.** *Given fixed target labels $\widetilde{\boldsymbol{Y}}_l$, the optimal nodes to poison are the subset of nodes corresponding to the smallest $\lfloor \epsilon L \rfloor$ negative elements of an $L$-dimensional vector $\boldsymbol{c}$, where the $l$-th element of $\boldsymbol{c}$ is computed as $c_l = \sum_{ij} Q_{il} P_{lj} R_{ij}$ where $\boldsymbol{Q} = \widehat{\boldsymbol{X}}_u \widetilde{\boldsymbol{X}}_l$, $\boldsymbol{P} = \widetilde{\boldsymbol{Y}}_l - \boldsymbol{Y}_l$, and $\boldsymbol{R} = \boldsymbol{Y}_u$.*

See § A for a proof. Intuitively, when $\tilde{\boldsymbol{Y}}_l$ is fixed we can rewrite the MILP in a canonical form as $\min_{\boldsymbol{b} \in \{0,1\}^L} \boldsymbol{c}^T \boldsymbol{b}$ subject to $\boldsymbol{b}^T \mathbf{1}_L \leq \epsilon L$ where $\boldsymbol{c}$ is a constant, which can be efficiently solved by returning the $\lfloor \epsilon L \rfloor$ smallest (negative) elements in $\boldsymbol{c}$.

## 4.2 META ATTACKS

For the second family of attacks we take an orthogonal approach where we directly optimize the poisoned labels $\widehat{\boldsymbol{Y}}_l$ via gradient descent. Since Eq. 1 is a bi-level problem we need to differentiate through the inner optimization. Here, unlike the previous attack **the inner problem is w.r.t. the actual target model** (e.g. the non-linear GCN) which is itself trained with gradient descent. Therefore, we resort to computing *meta gradients* by unrolling the inner optimization for a fixed number of epochs and differentiating through it. Unrolled optimization is common in the meta learning literature, where the goal is e.g. to learn the optimal hyper-parameters (or the structure) of a model (Huang et al., 2020;

---

[3]We use numerically stable algorithms to compute $\widetilde{\boldsymbol{X}}$ without explicitly computing the matrix inverse.

Mujkanovic et al., 2022). Intuitively, one can think of $\widehat{Y}_l$ as "hyper-parameters" to be optimized for the inner problem. At a high level, most meta-attacks follow a similar procedure by unrolling the inner-level optimization for K steps during each iteration of the outer loop. However, the attacks differ precisely in the problem-specific instantiating of this recipe. We explore three different ways (standard, fixed, expanded) to parameterize the poisoned labels.

**Standard meta attack.** We first initialize the free parameter matrix $\widetilde{Y}_{\log} \in \mathbb{R}^{L \times C}$ to learn the log-probability of flipping to a given target label for each training node. We obtain $\widetilde{Y}$ by sampling proportional to the log-probabilities in $\widetilde{Y}_{\log}$ using the Gumbel-softmax trick (Jang et al., 2017) on each row. In the forward pass $\widetilde{Y}_{\log}$ is "hard" (one-hot encoding), while using the softmax probabilities in the backward pass. We also initialize a parameter vector $\tilde{b} \in \mathbb{R}^L$ with the goal of learning which subset of nodes are a good candidate for poisoning. To enforce the budget we apply *soft-top-k* followed by *k-subset-selection* (Paulus et al., 2020) to obtain $b = \text{top}_k(\tilde{b})$. We construct the final poisoned labels by $H = \text{diag}(b)\widetilde{Y} + \text{diag}(1_L - b)Y_l$. We feed $H$ into the inner optimization problem which we train for a fixed number of epochs, and use meta-gradients of $b$ w.r.t. the outer loss for learning. During the forward pass we used hard top-$k$ for $b$, sampled proportional to the respective soft-top-$k$ scores which are used during the backward pass.

**Gumbel-softmax loss.** The choice of loss function in the outer problem can impact the attack performance. The loss of interest is the 0-1 loss (i.e. the test accuracy), which is non-differentiable. There are several options to circumvent this issue by constructing different approximations. The simplest idea is to just use the cross-entropy loss (which is also used in the inner problem) on the "soft" predictions $\widehat{Y}_l$. Another approach which was shown to perform better for other attacks (see e.g. (Mujkanovic et al., 2022)) is the margin loss, where we minimize the margin between the true class and the most probable false class. We propose a third alternative that exploits the Gumbel-Softmax trick. Specifically, in the forward pass we sample hard predictions from $\widehat{Y}_u$ using the Gumbel-Softmax reparametrization trick, while using the soft predictions for the backward pass. That is for node $i$ we sample a prediction via $\text{onehot}(\arg\max_c\{g_c + p_{ic}\})$ where $g_c \sim \text{Gumbel}(0,1)$ and $p_{ic}$ is the predicted probability for class $c$ obtained from the $(i,c)$-th entry of $\widehat{Y}_u$. Given the one-hot encoded hard predictions, we can compute the accuracy via the Hadamard product with the ground-truth $Y_u$. We study the choice of loss functions and their effects in B.8.2, since our newly proposed loss performs the best on average we use it in the main experiments.

**Fixed and expanded meta attacks.** We construct a fixed one-hot $\widetilde{Y} \in \{0,1\}^{L \times C}$ similar to our LSA attack by determining a fixed target label for each training node (e.g. the most likely false class for a clean model). Similar to the standard meta attack, we let $b$ learn to select the subset of poisoned labels, which now always flip to a fixed target label. We discuss the expanded variant in § B.3

**Adaptive attacks.** Our meta attacks are an instance of, so-called, adaptive attacks which were shown to be significantly better than surrogate-based attacks. This bitter lesson was learned for both computer vision (Carlini & Wagner, 2017; Athalye et al., 2018; Tramer et al., 2020) and GNNs (Mujkanovic et al., 2022). As we will see in § 5, adaptive attacks are not always superior for label poisoning.

### 4.3 The curious phenomenon of binary flips

We explore additional variants of all of our attacks, where instead of allowing the attacker to select the poisoned labels from the entire training set, we restrict their access only to a subset of labels corresponding to two classes. We refer to these as -BIN variants. This will allows us to compare the the prior baselines on their own terms since they do not natively support multi-class attacks. Similar to LAFAK we choose 2 candidate classes and always flip between them, e.g. the two most frequent classes. In § 5 we see that in practice the binary variants perform surprisingly well, often even better than their multi-class variants. This is interesting, since in theory any feasible solution for the binary variants is also feasible for the multi-class variants, i.e. multi-class is strictly more powerful. Even the RND and DEG baselines benefit. We investigate this phenomenon experimentally and theoretically.

**Surrogate loss gap.** We first identify a discrepancy between the binary and multi-class performance caused by the use of a least squares surrogate loss for the inner problem. We compute the optimal attacks and compare models trained with least squares (LS) vs. cross-entropy (CE). Then we compute the difference in accuracy when we switch from LS (the attack's surrogate) to CE (the defender's

Table 1: Difference in performance when training the model with cross-entropy vs. least squares loss for Cora-ML. The multi-class attack has significantly higher $\Delta$ in accuracy, overfitting to the linear regression surrogate.

| $\epsilon$ | $\Delta_{\text{acc}}^{\text{bin}}$ | $\Delta_{\text{acc}}^{\text{mul}}$ | $\Delta_{\text{loss}}^{\text{bin}}$ | $\Delta_{\text{loss}}^{\text{mul}}$ |
|---|---|---|---|---|
| **5%** | -0.02 | +1.79 | +0.76 | +0.78 |
| **10%** | +1.55 | +3.76 | +0.86 | +0.88 |
| **15%** | +0.69 | +6.01 | +0.95 | +0.96 |
| **20%** | -0.82 | +8.36 | +1.09 | +1.06 |
| **30%** | -2.18 | +9.88 | +1.33 | +1.27 |

Table 2: Accuracy comparison between multi-class and binary SGC attacks *without* hyperparameter tuning, on Cora-ML and GCN. The multi-class attack is stronger here, but not after tuning (see Fig. 3) indicating overfitting.

| $\epsilon$ | **multi** | **binary** |
|---|---|---|
| **5%** | 74.49 (2.22) | 75.01 (1.13) |
| **10%** | 67.02 (2.43) | 70.35 (1.80) |
| **15%** | 59.10 (1.81) | 65.50 (1.86) |
| **20%** | 54.76 (1.46) | 60.08 (3.04) |
| **30%** | 47.76 (1.45) | 45.78 (4.20) |

actual evaluation) denoted with $\Delta_{\text{acc}}^{\text{bin}}$ for the binary attack (and $\Delta_{\text{acc}}^{\text{mul}}$ for the multi-class attack). In Table 1 we see that difference in accuracy is negligible when using the binary attack, and quite large for the multi-class attack, even growing with increased budget $\epsilon$. The loss shows no significant difference. Thus, the binary variants perform better since they incur a smaller surrogate loss gap.

**Adversarial overfitting.** We also identify overfitting to the default h-parameters. In Table 2 we see that when we run the SGC attack on an *untuned* model, the multi-class variant is better, however, this is not true for the tuned model (see Fig. 3). Both findings indicate that the *unconstrained* multi-class attacks are more likely to overfit – to either the surrogate or the untuned model – and thus are unable to find poisons that generalize to the final tuned model used by the defender. Binary attacks are "regularized". Finally, for the random heuristic attacks we can prove that the binary variant is optimal.

**Proposition 3.** *Let the adversary flip label $p$ to label $q \neq p$ with probability $\frac{\epsilon}{s} \cdot t_{pq}$ and retain label $p$ with probability $1 - \epsilon$, where $\epsilon$ is the poisoning budget, $t_{pq} \in \{0, 1\}$ indicates whether the adversary is allowed to flip $p$ to $q$, and $s = \sum_{q \neq p} t_{pq}$ is the number of allowed classes. The test accuracy of the Bayes optimal classifier trained on randomly flipped labels is minimized for $s = 1$ (binary flips).*

## 5 EXPERIMENTAL EVALUATION

We conduct a comprehensive comparison between our family of attacks and all baseline methods. Furthermore, for the first time and in response to one of the identified pitfalls, we assess our attacks against two *defense mechanisms* specifically designed to withstand label perturbations: CPGCN (Zhang et al., 2020) which introduces a clustering-based loss, and RTGNN (Qian et al.), which uses self-reinforcement and consistency regularization. Label poisoning can also be viewed as an extreme version of feature perturbation. To account for this, we also evaluate our strongest attack and the strongest baseline (LFK) against ALBATIONGCN (Scholten et al., 2022), one of the recent certified defenses against feature perturbations for GNNs (see § B.16). See § B for implementation and tuning details. In § B.8 we provide additional results, including larger scale graphs and ablations. The change in homophily due to poisoning is minor (§ B.11). In all experiments, we use the final evaluation setting (CV) that we propose in § 3 which avoids all of the identified pitfalls.

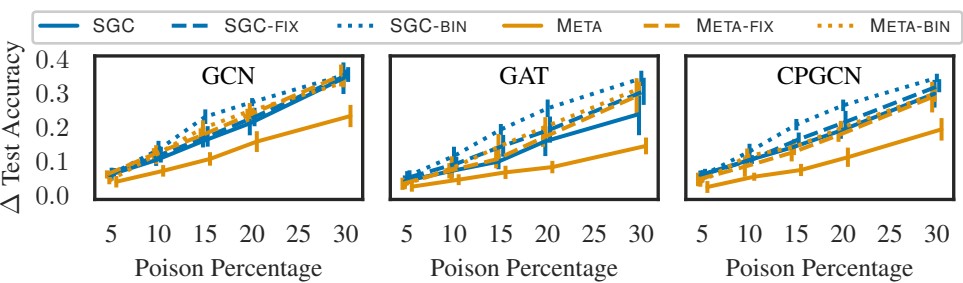

Figure 3: Different variants of the attacks that we propose. Binary variants are best on average.

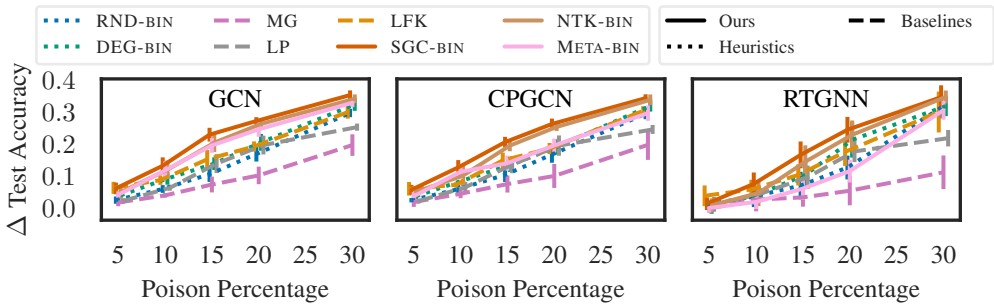

Figure 4: Our strongest attacks significantly outperform the baselines across various models.

**Attack variants.** We first study the different variants of our attacks. We denote with -FIX and -BIN the fixed and binary subset variants respectively. In Fig. 3, we show the difference in the accuracy of clean vs. poisoned labels for different budgets. We infer that on average linear surrogate attacks are better than meta-based attacks. One reason is the fact that we are able to solve the MILP in Eq. 2 exactly (and efficiently). That is, we find the optimal attack (given the surrogate), while the meta-based attacks suffer from optimization issues. Binary variants are significantly stronger than multi-class. Similar trends hold for the baselines (see § B.8). We discussed this phenomenon in § 4.3.

**Attacks.** For ease of comparison, we pick the best performing (on average) attack from each family that we introduced in § 4, namely: SGC-BIN, NTK-BIN and META-BIN. We compute the difference in test performance between the clean-accuracy and poisoned accuracy for each budget across attacks, and plot them on Fig. 4. Higher difference shows a stronger attack. We observe that all our attacks (in solid), particularly SGC-BIN attack, outperform baseline attacks with maximum gains of up to $\sim 13\%$. We observe similar trends across different models and datasets as we show in § B.8.

CPGCN and RTGNN are two recent defenses that have been introduced to specifically tackle pairwise noise in labels – where the flips between classes is fixed. Attacks like LFK, LP, and the binary variants of our proposed attacks can be viewed as (*adversarial*) pairwise noise. While RTGNN has claimed impressive recovery in performance with pairwise noise rate of up to 40%, we find that our simple attacks can cause a debilitating effect on its performance with as low as 10% noise in labels.

Surprisingly the SGC-based attacks transfer well to a wide range of vanilla GNNs as well as robust GNNs. Moreover, our simple attack formulation significantly outperforms prior attacks that use complicated routines. In contrast to the success of meta-gradient attacks for graph/feature perturbations (Mujkanovic et al., 2022), we observe meta attacks to be inferior on average.

**Ablation and limitations.** We further examine how each attack performs per split, and also qualitatively analyse the effect of poisoning on the learned representations. These results are available in § B.7. Moreover, our attacks are unsurprisingly even stronger on the default (flawed) setting as we show in § B.12. While it may not always be easy to implement our threat models in practice, we still gain significant insights into the inherent resilience of the models. Our theoretical insights (Proposition 3) apply only to random flips, and extending this to the worst-case is left for future work.

## 6 DISCUSSION AND CONCLUSION

There are two main messages in our work. First, careful evaluation of poisoning attacks is critical to accurately assess robustness. All examined previous studies commit serious pitfalls for which we provide remedies. The crux of the issue is faithfully simulating the defender when testing an attack, which includes proper tuning and splitting strategies. Second, simple label poisoning attacks are surprisingly powerful – all examined models, including those designed to be robust, are highly vulnerable. Interestingly, contrary to popular conceptions, in our setting linear surrogate attacks outperform adaptive meta attacks. We gave experimental and theoretical insights into a curious phenomenon of why binary variants perform better compared to their multi-class counterpart. It would be interesting to study this in more depth in the future. Our findings highlight the urgent need to further study poisoning attacks, as well as develop robust defenses and certificates against them.

## REPRODUCIBILITY STATEMENT

We offer comprehensive technical descriptions of our proposed attacks in § 4. The hyper-parameter ranges we explore for both the attack and defense phases are included in the initial section of § B. We provide proofs for our propositions in § A. Furthermore, we make our code publicly available.

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

## A    PROOFS

When the target labels are fixed we can efficiently obtain the optimal subset of poisoned nodes in a closed-form solution with Proposition 2. However, even when we optimize over $H$, i.e. optimize over both the subset and the target labels, we can still solve the MILP efficiently. We will show that relaxing $H$ from $\{0,1\}^{L \times C}$ to $[0,1]^{L \times C}$ in Eq. 2 still results in an integral (i.e. binary) solution. Therefore, we can solve the MILP by solving the corresponding relaxed LP.

*Proof (Proposition 1).* Let $M = N \cdot C$. With $\mathcal{L}(\boldsymbol{Y}_u, \widehat{\boldsymbol{Y}}_u) = \text{sum}(\boldsymbol{Y}_u \odot \widehat{\boldsymbol{Y}}_u)$ we can rewrite Eq. 2 in the following canonical form

$$\min_{\boldsymbol{x} \in \{0,1\}^{2 \cdot M}} \boldsymbol{c}^T \boldsymbol{x}$$
$$\boldsymbol{A}\boldsymbol{x} = \boldsymbol{b}$$
$$\boldsymbol{G}\boldsymbol{x} \leq \boldsymbol{h}$$

where $\boldsymbol{A} = [\underbrace{\boldsymbol{I}_N, \ldots, \boldsymbol{I}_N}_{C \text{ times}}, \boldsymbol{O}]$, $\boldsymbol{I}_N$ is the identify matrix, $\boldsymbol{O}$ is an $N \times M$ zero matrix, $\boldsymbol{b} = \boldsymbol{1}_N$ is the

ones vector, $\boldsymbol{G} = \begin{bmatrix} -\boldsymbol{I}_M & -\boldsymbol{I}_M \\ \boldsymbol{I}_M & -\boldsymbol{I}_M \\ \boldsymbol{0}_M & \boldsymbol{1}_M \end{bmatrix}$, $\boldsymbol{0}_M$ is the zeros vector, and $\boldsymbol{h} = [-\text{vec}(\boldsymbol{Y}_l), \text{vec}(\boldsymbol{Y}_l), \epsilon 2N]$, and

$\boldsymbol{c} = \text{vec}(\boldsymbol{Y}_u^T \widehat{\boldsymbol{X}}_u \widetilde{\boldsymbol{X}}_l)$. Here, $\boldsymbol{A}\boldsymbol{x} = \boldsymbol{b}$ implements the one-hot constraint, while $\boldsymbol{G}\boldsymbol{x} \leq \boldsymbol{h}$ implements the $L_1$-norm budget constraint, where we have twice the number of variables to encode the absolute value. Since $\boldsymbol{b}$ and $\boldsymbol{h}$ are integral (because $\boldsymbol{Y}_l$ is one-hot encoded), to show that the optimal solution is integral we only need to show that $\boldsymbol{A}$ and $\boldsymbol{G}$ are totally unimodular. Since the identity matrix $\boldsymbol{I}_N$ is totally unimodular, the result directly follows given the fact that if $\boldsymbol{T}$ is totally unimodular then so are $-\boldsymbol{T}, \boldsymbol{T}^T, [\boldsymbol{T}, \boldsymbol{I}]$, and $[\boldsymbol{T}, -\boldsymbol{T}]$. $\qquad\square$

*Proof (Proposition 2).* When the target labels labels are fixed we can rewrite the MILP from Eq. 2 in terms of auxiliary variables $\boldsymbol{b}$ that control the subset of poisoned labels. The resulting MILP is:

$$\min_{\boldsymbol{b} \in \{0,1\}^L} \mathcal{L}(\boldsymbol{Y}_u, \widehat{\boldsymbol{Y}}_u) \tag{3a}$$
$$\boldsymbol{H} = \text{diag}(\boldsymbol{b}) \odot \widetilde{\boldsymbol{Y}}_l + \text{diag}(\boldsymbol{1}_L - \boldsymbol{b}) \odot \boldsymbol{Y}_l \tag{3b}$$
$$\widetilde{\boldsymbol{Y}}_u = \widehat{\boldsymbol{X}}_u \widetilde{\boldsymbol{X}}_l \boldsymbol{H} \tag{3c}$$
$$\boldsymbol{b}^T \boldsymbol{1}_L \leq \epsilon L \tag{3d}$$

Given $\mathcal{L}(\boldsymbol{Y}_u, \widehat{\boldsymbol{Y}}_u) = \text{sum}(\boldsymbol{Y}_u \odot \widehat{\boldsymbol{Y}}_u)$ we can rewrite the MILP in Eq. 3 (and equivalently Eq. 2) as

$$\underset{\boldsymbol{b} \in \{0,1\}^N}{\arg\min} \, \text{vec}(\boldsymbol{Y}_u)^T \text{vec}\left(\widehat{\boldsymbol{X}}_u \widetilde{\boldsymbol{X}}_l \boldsymbol{H}\right) \qquad \boldsymbol{b}^T \boldsymbol{1}_N \leq \epsilon N \tag{4}$$

We can simplify Eq. 4 without changing the minimum by omitting the terms that do not depend on $\boldsymbol{b}$:

$$\underset{\boldsymbol{b} \in \{0,1\}^N}{\arg\min} \, \text{vec}(\boldsymbol{Y}_u)^T \text{vec}\left(\widehat{\boldsymbol{X}}_u \widetilde{\boldsymbol{X}}_l \left(\text{diag}(\boldsymbol{b}) \odot (\widetilde{\boldsymbol{Y}}_l - \boldsymbol{Y}_l)\right)\right) \qquad \boldsymbol{b}^T \boldsymbol{1}_N \leq \epsilon N \tag{5}$$

Now if we define $\boldsymbol{Q} = \widehat{\boldsymbol{X}}_u \widetilde{\boldsymbol{X}}_l$, $\boldsymbol{P} = \widetilde{\boldsymbol{Y}}_l - \boldsymbol{Y}_l$ and $\boldsymbol{R} = \boldsymbol{Y}_u$ we can again rewrite as:

$$\underset{\boldsymbol{b} \in \{0,1\}^N}{\arg\min} \, \boldsymbol{c}^T \boldsymbol{b} \qquad \boldsymbol{b}^T \boldsymbol{1}_N \leq \epsilon N$$

where the $l$-th element of $\boldsymbol{c}$ is computed as $c_l = \sum_{ij} Q_{il} P_{lj} R_{ij}$. This follows since the objective in Eq. 5 equals $\sum_{ilj} (Q_{il} \cdot b_l \cdot P_{lj} \cdot R_{ij})$. This problem is equivalent to selecting up to $\epsilon N$ elements from $\boldsymbol{c}$ such that their sum is minimized. The optimal solution is to select the $\lfloor \epsilon N \rfloor$ smallest *negative* elements (or all negative elements if the number of negative elements is $< \lfloor \epsilon N \rfloor$). $\qquad\square$

*Proof (Proposition 3).* Our proof relies on the theory about robustness to label noise that was recently derived in Oyen et al. (2022). We denote with $Y^*$ the random variable corresponding to the ground-truth labels, and with $Y$ the random variable for the noisy labels. We define the distribution that randomly flips ground-truth $Y^*$ to noisy $Y$ as:

$$\eta_{pq} = \begin{cases} 1 - \epsilon & p = q \\ \frac{\epsilon}{s} t_{pq} & p \neq q \end{cases}$$

From the definition, $Y \mid Y^*$ is a multinomialy distributed and $\eta_{pq}$ determines which noisy (poisoned) label is chosen. Let $X$ denote the random variable for the features.

Denote with $f(\boldsymbol{x})_q = \text{Prob}[Y = q \mid X = \boldsymbol{x}] = \sum_{p=1}^{C} \eta_{pq} f^*(\boldsymbol{x})_q$ the *noisy* posterior of the Bayes optimal classifier, where $f^*(\boldsymbol{x})_q$ is the analogous *clean* (unperturbed) posterior. From the definition of $\eta_{pq}$ (and $t_{pq}$) we see that $s$ controls the number of allowed target classes, and $s = 1$ corresponds to binary flips. Technically, for $s = 1$ each class can choose a different (but single) target class. This is slightly more generally that our binary variants. For $s = C - 1$, where $C$ is the total number of classes, we recover the multi-class variant of the random heuristic baseline.

Oyen et al. (2022) showed that clean accuracy (for class $q$) depends on $s$:

$$\mathbb{E}[\mathbb{I}(\arg\max_i f(\boldsymbol{x})_i = q) \mid Y^* = q] \approx \frac{\bar{f}_q^*}{1 + \exp \frac{s+1}{s} \cdot (2\bar{f}_q^* - 1) \cdot (\epsilon - \frac{s}{s+1})} \triangleq r(s) \qquad (6)$$

where $\bar{f}_q^* = \mathbb{E}[f^*(\boldsymbol{x})_q]$. They also show that the noisy accuracy (for class $q$) depends on $s$:

$$E[\mathbb{I}(\arg\max_i f(\boldsymbol{x})_i = q) \mid Y = q] = \bar{f}_q^* \left(1 - \frac{\epsilon(s+1)}{s}\right)^2 + \frac{\epsilon}{s}\left(2 - \frac{\epsilon(s+1)}{s}\right) \qquad (7)$$

To prove Proposition 3 we need to show that the clean accuracy (for class $q$) in Eq. 6 as a function of $s$ is minimized for $s = 1$ for each class $q$. Analogously, one can easily show that Eq. 7 is also minimized for $s = 1$, but we focus on Eq. 6 since it is the more relevant quantity for our setting.

We will prove that the function $r(s)$ in Eq. 6 is monotonically increasing for $s = 1, 2, \ldots, C - 1$. First, we need to find its derivative w.r.t. $s$ and analyze its sign. We have:

$$\nabla r(s) = -\frac{\bar{f}_q^*}{s} \times \frac{e^{\left(\frac{(-1+2\bar{f}_q^*)(1+s)\left(\epsilon - \frac{s}{1+s}\right)}{s}\right)}}{s}$$
$$\times \left( \frac{(-1 + 2\bar{f}_q^*)(1 + s)\left(\frac{s}{(1+s)^2} - \frac{1}{1+s}\right)}{s} \right.$$
$$\left. + \frac{(-1 + 2\bar{f}_q^*)\left(\epsilon - \frac{s}{1+s}\right)}{s} - \frac{(-1 + 2\bar{f}_q^*)(1 + s)\left(\epsilon - \frac{s}{1+s}\right)}{s^2} \right) \qquad (8)$$
$$\times \frac{1}{\left(1 + e^{\left(\frac{(-1+2\bar{f}_q^*)(1+s)\left(\epsilon - \frac{s}{1+s}\right)}{s}\right)}\right)^2}$$

We need to analyze the sign of each component of this expression. Since, $0 \leq \bar{f}_q^* \leq 1$ the first term is non-negative. The second term is always positive due to the exponential function. The last term is positive due to the square in the denominator.

It remains to analyze the sign of the third term inside the largest parentheses. This term expands to:

$$\frac{\epsilon}{s^2} - \frac{2\bar{f}_q^*\epsilon}{s^2} - \frac{1}{(1+s)^2} + \frac{2\bar{f}_q^*}{(1+s)^2} - \frac{s}{(1+s)^2} + \frac{2\bar{f}_q^*s}{(1+s)^2} + \frac{1}{1+s} - \frac{2\bar{f}_q^*}{1+s}$$

With some algebra we show that the term simplifies to:

$$\frac{(\epsilon - 2\bar{f}_q^*\epsilon)}{s^2} \qquad (9)$$

This means that the sign of the entire expression is determined by the sign of Eq. 9. First, we consider the case that $\frac{1}{s} < \bar{f}_q^* \leq 1$. Since $s$ is positive we only care about the nominator which we can rearrange as $\epsilon(1 - 2\bar{f}_q^*)$. Since $0 < \epsilon < 1$ is positive we need to show $1 - 2\bar{f}_q^* < 0$. Given that $\bar{f}_q^* > \frac{1}{s}$, we have that $2\bar{f}_q^* > \frac{2}{s}$. This implies that $2\bar{f}_q^* > 1$ since $s$ is positive and $s = 1$ in the worst-case. Which means $1 - 2\bar{f}_q^* < 0$.

Now, combining all the components: the negative sign in front of the derivative and the negative term $\frac{(\epsilon - 2\bar{f}_q^* c)}{s^2}$ together make the derivative $\nabla r(s)$ positive for all $s = 1, \ldots, C - 1$.

It remains to consider the case where $\bar{f}_q^* \leq \frac{1}{s}$. Similar to Oyen et al. (2022) we argue that in this case the classification is no better than random chance which would give a vacuous result since in this case label noise actually would improve (rather than hurt) the performance.

$\square$

## B  APPENDIX

The appendix is structured as follows. § B.1 contains additional details on the motivation experiment we introduced in the main paper. In § B.2, we provide an extended discussion of related work. In § B.3, we provide details on the expanded meta attack. § B.4 contains details regarding our NTK-based attacks. In § B.5, we provide additional details on the variants of our LSA attack. In § B.6, we provide details on extending the baselines to multi-class. In § B.7, we provide qualitative and quantitative analysis of poisoning attacks. In § B.8, we provide additional experiments along with ablative studies to offer more insights. § B.9 provides details on dataset statistics. In § B.10, we show how label poisoning attacks can be thwarted if a clean validation dataset is assumed to be available (unrealistic setting). § B.11 discusses the effect of label poisoning on graph homophily. In § B.12, we evaluate our attacks on the previously used (flawed) setting. In § B.13, we study the effectiveness of our attacks under partial information scenario. § B.13 provides timing analysis for our proposed evaluation setting. § B.15 provides label frequency analysis for our benchmark datasets. In § B.16, we evaluate our attacks against a certified defense model. In § B.17, we provide additional ablation studies on pitfalls. Finally, in § B.18, we provide insights on the connections between the identified pitfalls and the attacks we devised.

First, we discuss how we tune the hyper-parameters in both the attack and the defense phase of our evaluation.

**Attack phase tuning.** For our LSA family of attacks, we fine-tuned the regularisation term $\lambda$ within the range [1e-3, 1e3] in logarithmic steps. For the meta attacks, we set the learning rate for the inner and the outer routine optimization to 1e-2 and 1e-1. The weight decay for both the optimizers are set to 5e-4. The number of inner optimization iterations (epochs) is set to 15. Additional gains in the attack strength can be achieved by further fine-tuning these hyper-parameters. We refrain from doing this due to lack of computational resources, since our evaluation setup is already quite expensive. For the learning-based baseline attacks (LP, MG, LFK), we sweep through the hyper-parameters ranges suggested by the authors in their papers.

**Defence phase tuning.** Post obtaining the poisoned labels, as identified in our of pitfalls § 3, we perform thorough hyper-parameter tuning of the target model. Specifically, for the shared general hyper-parameters we sweep the following ranges: [0.1, 0.01, 0.05, 0.08] for the learning rate, [0.0, 0.005, 0.0005, 0.00005] for weight decay, and [0.3, 0.5, 0.7] for dropout. We fix the hidden dimensions to 64, again due to limited computation resources. Model specific hyper-parameters are tuned in the ranges suggested by the respective papers. We empirically observe that higher learning rates allow models to recover better from label poisoning attacks. We use Optuna (Akiba et al., 2019) to optimize the hyper-parameters search, and set the number of trials to 20. For learning we use the Adam optimizer (Kingma & Ba, 2015), with 1000 maximum number of epochs and an early stopping patience of 100. The majority of our experiments are run on an Nvidia k80 GPU with 24GB memory, and the remaining experiments were run using an A100 GPU with 40GB of memory.

### B.1 MOTIVATION EXPERIMENT

In the motivation experiment discussed in the main paper, we focused on the Cora-ML dataset and extracted a sub-sample from its largest connected component. This sub-sample is referred to as Cora-ML-tiny, and consists 93 nodes categorized into 3 classes. To conduct our analysis, we created 10 distinct splits for training, validation, and test sets, each with a size of 19, 19, and 55, respectively. By exhaustively examining all possible label flips within a budget constraint of 1, we discovered that a single adversarial label flip could lead to a significant reduction of ∼17% in test performance on average.

### B.2 EXTENDED DISCUSSION OF RELATED WORK

Adversarial attacks for machine learning (ML) is an active and important area of research, especially with ML applications becoming ubiquitous, and have been studied extensively in the literature (Jin et al., 2021; Sun et al., 2023; Liang et al., 2022). Attacks can be broadly classified into *evasion* and *poisoning*. For evasion attacks the model is fixed and the attacker perturbs the input at test time. Evasion attacks are conducted during the testing phase, where the model is fixed and the attacker perturbs a test instance to degrade model performance (Biggio & Roli, 2018). Poisoning attacks are conducted *before* the model is trained, where the attacker can perturb the training data with an intention to degrade model performance or control model behavior. Poisoning is significantly more challenging than evasion since it results in a difficult bi-level optimization problem.

**Robustness of GNNs.** Currently, we have an extensive body of work on the (adversarial) robustness of GNNs. The overwhelming majority of studies focus on adversarial attacks that add or remove a small fraction of edges and/or perturb the node features. Consequently, various heuristic defenses have been developed, as well as certificates to provide provable robustness guarantees. For a detailed overview of both attacks and defenses we recommend the surveys by Sun et al. (2023), Chen et al. (2020), and Jin et al. (2021)). Label poisoning attacks, despite their importance and potential impact, have been surprisingly sparse with only a few exceptions (Liu et al., 2019; Zhang et al., 2020; Liu et al., 2022). Interestingly, Mujkanovic et al. (2022) recently showed that adaptive attacks – designed specifically to circumvent a given target model or defense – are still able to manipulate almost all models. In other words, we have made significantly less progress than it initially appears towards designing robust models. One conclusion from their work is that we have to carefully think about the evaluation setup to avoid overly optimistic results. This is echoed in our pitfalls findings.

**Poisoning attacks.** Most poisoning attack assume that the attacker has access to the training data, and position the proposed attack as a worst-case robustness analysis. Such attacks have been extensively studied for classical ML models like SVMs (Biggio et al., 2011), as well as recent (deep learning) models. Lots of different threat models have been considered, including allowing arbitrary changes (both features and labels), clean label poisoning (only features), and poisoning only the labels, which is our focus. See Jagielski et al. (2018) and Tian et al. (2022) for a comprehensive overview.

### B.3 EXPANDED META ATTACK

We first construct $\widetilde{\boldsymbol{Y}} \in \{0,1\}^{N \times C-1 \times C}$ by enumerating over all possible false (one-hot) labels for every node in the training set. Next, we initialize a parameter matrix $\boldsymbol{B} \in \mathbb{R}^{N \times C-1 \times C}$ from a uniform distribution. The idea is to learn $\boldsymbol{B}$ to select a good subset of poisoned labels. To enforce the budget we apply *soft-top-k* followed by *k-subset-selection* (Paulus et al., 2020) on top of $\mathrm{vec}(\boldsymbol{B})$ where $\mathrm{vec}(\cdot)$ vectorizes the matrix.[4] We construct the final poisoned labels as:

$$\boldsymbol{H} = \sum_{c=0}^{C-1} (\boldsymbol{B} \odot \widetilde{\boldsymbol{Y}})_{:,c,:} + ((1-\boldsymbol{B}) \odot \boldsymbol{Y}_l)_{:,c,:} \tag{10}$$

We feed $\widetilde{\boldsymbol{Y}}$ into the inner optimization problem which we train for a fixed number of epochs, and use meta-gradients of $\boldsymbol{B}$ w.r.t. the outer loss for learning. During the forward pass we used hard top-$k$ for $\boldsymbol{B}$, sampled proportional to the respective soft-top-$k$ scores which are used during the backward pass.

---

[4]We place no additional constraints on $\boldsymbol{B}$ to ensure multiple flips are not selected for the same label. In practice this is not an issue, since the optimizer learns to use the entire budget and select $k$-unique label flips.

## B.4 NTK ATTACK

Our variants of the NTK-based attacks are exactly the same as our SGC variants, except in how we construct $\widehat{X}$. For the NTK variants, $\widehat{X}$ corresponds to the NTK kernel matrix for a 2 layer ReLU GCN as derived by Sabanayagam et al. (2022) (see Eq. 5 in their paper). Given $\widehat{X}$ we generate poisoned labels following the procedure in § 4.1.

## B.5 ADDITIONAL DETAILS ON THE MILP FORMULATION OF LSA

In one variation of our LSA attacks instead of learning $H$, we construct it by selecting a subset of nodes to poison from preset target false labels $\widetilde{Y}_l$.

As we showed in Proposition 2 the benefit of fixing $\widetilde{Y}$ is that we can obtain a closed-form solution for the MILP. We explore the following variants:

- $\widetilde{Y}$-MARGIN: we perturb labels to the false label with the highest probability as predicted by a clean model (trained using ground truth training labels).
- $\widetilde{Y}$-RANDOM: we perturb labels to a randomly chosen false class.
- $\widetilde{Y}$-BINARY+: we sort all classes w.r.t their frequency in a descending order and pair consecutive classes to flip them.

We compare these variants for $\widetilde{Y}$ in § B.8. In the main text we use $\widetilde{Y}$-MARGIN for the -FIX attacks.

## B.6 EXTENDING THE BASELINES TO MULTI-CLASS

As a consequence of the binary-class approach, the baselines might have a limit on the maximum perturbation budget. For example, in the default (previously-used, non-CV) evaluation setting for the Cora-ML dataset the candidate set of potential flips is of size at most 20% since Cora-ML has 7 classes and the two most frequent classes span up to 20% of the training set depending on the split. Therefore, LFK and LP by default cannot accommodate higher budgets. To enable this, while not deviating from the original design of the attack, we propose a minor extension. We first exhaust the 20% budget by perturbing labels of the two most frequent classes and then we fix these perturbed labels. For the remaining budget, we perturb the clean labels, but by restricting the attack scope to a candidate set consisting of the next two most frequent classes. This process is repeated until the budget is completely exhausted. The multi-class setting can be handled better, however, we restrain from such adaptations to remain true to the original design of the attack. This also means that for the 20% budget, in the previous evaluation setting, LFK and LP become equivalent and deterministic because all the labels of the nodes in the candidate set are flipped to their counterpart. Note, that this issue is *not present* once we apply our remedies from § 3, and is only relevant for the experiments showing the pitfalls of previous evaluations.

## B.7 QUANTITATIVE AND QUALITATIVE ANALYSIS OF POISONING ATTACKS

On Fig. 5 we show how many times a given attack wins, i.e. leads to lowest accuracy, across all splits and all models (for Cora-ML and Citeseer). Different variants of our linear attacks win most often.

**Analysing the poison.** Finally, we study how the learned representations change before and after poisoning with SGC-BIN. On Fig. 6 (left) we see that for the clean model the representations cluster nicely according to the class shown with different colors. We also see that the subset of poisoned nodes (shown with a cross) is spread out. The poisoned representations (right) for different classes are highly overlapping. Moreover, the two target classes that we flip (green and pink) overlap the most.

## B.8 ADDITIONAL EXPERIMENTS

### B.8.1 ADDITIONAL EXPERIMENTS ON STRONGEST ATTACKS

To extend our main paper analysis on strongest attacks, we provide additional experimental results for our strongest attacks ( SGC-BIN and META-BIN) on several datasets and models. We plot the

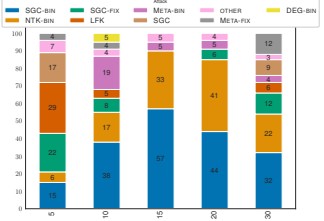

Figure 5: Strongest per split.

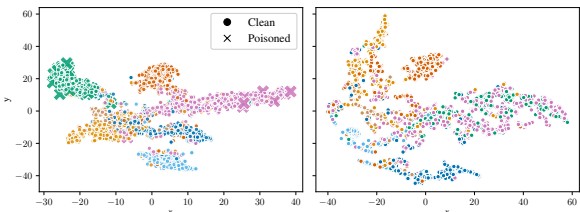

Figure 6: t-SNE of the logits before and after poisoning.

results in Fig. 9. Our findings from the main paper that our attacks significantly outperform previous attacks still hold.

### B.8.2 EFFECT OF LOSS FUNCTIONS ON PROPOSED META ATTACKS

We study the effect of different loss functions – Cross Entropy, Margin, and the proposed Gumbel Softmax loss, on the META attack for the Cora-ML dataset. We tabulate the results in Table 3. We can observe that the proposed Gumbel Softmax based loss function boosts the attack performance in general.

Table 3: Effect of loss functions for the proposed META attack on the Cora-ML dataset.

| Poison Ratio | Cross Entropy Loss | Margin Loss | Gumbel Softmax Loss |
|:---:|:---:|:---:|:---:|
| 5% | $\mathbf{78.51_{\pm 2.18}}$ | $79.25_{\pm 1.68}$ | $79.80_{\pm 2.12}$ |
| 10% | $75.96_{\pm 1.39}$ | $75.27_{\pm 1.93}$ | $\mathbf{73.35_{\pm 2.34}}$ |
| 15% | $67.78_{\pm 1.81}$ | $68.75_{\pm 3.20}$ | $\mathbf{65.81_{\pm 3.30}}$ |
| 20% | $63.36_{\pm 3.56}$ | $63.88_{\pm 2.90}$ | $\mathbf{61.60_{\pm 3.37}}$ |
| 30% | $56.13_{\pm 3.05}$ | $56.17_{\pm 2.74}$ | $\mathbf{54.53_{\pm 2.28}}$ |

### B.8.3 SCALING OUR ATTACKS TO LARGER GRAPHS AND LARGER NUMBER OF CLASSES

To further study the efficacy of our proposed attacks, we extend our analysis to two additional datasets. Specifically, we evaluate on the CoraFull dataset (Bojchevski & Günnemann, 2018) that contains 70 output classes, and the OGBN-ArXiv dataset (Hu et al., 2020) – a medium sized graph with $\sim$ 169K nodes. Following Lingam et al. (2023), we apply a dimensionality reduction on the node features of CoraFull using PCA (top 500 dimensions). The statistics for all 5 datasets are tabulated in Table 7.

**Larger number of classes.** Since the number of classes for CoraFull is relatively higher compared to the other datasets, the choice of how we fix $\widetilde{Y}$ can be potentially more impactful. Therefore, we compare the three variants outlined in § B.5 against the baselines. We can infer from Table 4 that our attacks outperforms existing attacks, and that the -BINARY+ variant is the best on average. These results signify that our attacks also scale well to datasets with larger number of classes.

Table 4: Comparing different attacks on the CoraFull dataset. The strategy for how to initialize $\widetilde{Y}$ can impact the attack performance. The -MARGIN and -BINARY+ variants outperform -RANDOM.

| $\epsilon$ | $\widetilde{Y}$-MARGIN | $\widetilde{Y}$-BINARY+ | $\widetilde{Y}$-RANDOM | DEG | RND | LP | MG |
|:---|:---:|:---:|:---:|:---:|:---:|:---:|:---:|
| 5% | $56.14_{\pm 0.53}$ | $\mathbf{55.96_{\pm 0.61}}$ | $57.41_{\pm 0.52}$ | $\mathbf{55.70_{\pm 0.42}}$ | $57.01_{\pm 0.45}$ | $57.38_{\pm 0.26}$ | $56.19_{\pm 0.53}$ |
| 10% | $51.27_{\pm 0.54}$ | $\mathbf{50.22_{\pm 0.84}}$ | $54.49_{\pm 0.50}$ | $54.02_{\pm 0.62}$ | $55.21_{\pm 0.59}$ | $53.52_{\pm 0.27}$ | $52.15_{\pm 0.48}$ |
| 15% | $46.31_{\pm 0.57}$ | $\mathbf{43.99_{\pm 0.27}}$ | $51.67_{\pm 0.69}$ | $52.86_{\pm 0.66}$ | $54.74_{\pm 0.73}$ | $49.66_{\pm 0.12}$ | $47.58_{\pm 0.53}$ |
| 20% | $41.47_{\pm 0.28}$ | $\mathbf{38.10_{\pm 0.54}}$ | $48.30_{\pm 0.93}$ | $51.53_{\pm 0.71}$ | $53.56_{\pm 0.52}$ | $49.08_{\pm 0.08}$ | $43.02_{\pm 0.56}$ |
| 30% | $33.29_{\pm 0.68}$ | $\mathbf{27.92_{\pm 0.30}}$ | $40.93_{\pm 1.60}$ | $48.56_{\pm 0.86}$ | $51.29_{\pm 0.75}$ | $49.10_{\pm 0.10}$ | $31.71_{\pm 0.08}$ |

Table 5: Comparing different attacks on OGBN-ArXiv. Our SGC-BIN attack also scales to large graphs and still outperforms existing attacks by a significant margin. The LFK attack, with its current implementation, could not be executed because of Time Limit Exceeded error.

| $\epsilon$ | SGC-BIN | RND | DEG | LP | MG | LFK |
|---|---|---|---|---|---|---|
| 1% | **64.31** | 67.14 | 66.56 | 66.15 | 67.21 | TLE |
| 3% | **62.17** | 67.09 | 66.36 | 60.56 | 66.69 | TLE |
| 5% | **48.20** | 66.90 | 66.12 | 55.62 | 66.72 | TLE |

Table 6: Evaluation of the extended variant of our meta attack on a GCN model for different datasets.

| $\epsilon$ | Cora-ML | Citeseer | Pubmed |
|---|---|---|---|
| 5% | $80.68_{\pm1.65}$ | $70.62_{\pm1.31}$ | $76.15_{\pm1.34}$ |
| 10% | $79.57_{\pm1.86}$ | $68.85_{\pm0.73}$ | $73.20_{\pm1.97}$ |
| 15% | $77.76_{\pm1.30}$ | $66.96_{\pm1.80}$ | $69.02_{\pm3.60}$ |
| 20% | $75.34_{\pm1.74}$ | $64.24_{\pm3.03}$ | $61.28_{\pm3.86}$ |
| 30% | $71.73_{\pm2.13}$ | $60.08_{\pm3.41}$ | $50.51_{\pm6.71}$ |

**Larger graphs.** Next, we study the medium-sized OGBN-ArXiv dataset to test scalability to larger graphs. Since the OGBN-ArXiv dataset contains large number of training nodes, we evaluate the attacks on budget ranges {1%, 3%, 5%}. We also use the default splits since the data is chronologically split. In Table 5 we see that our proposed attack, SGC-BIN, outperforms existing attacks.

### B.8.4 CONFUSION MATRIX ANALYSIS

To analyse how the test node predictions are affected by the strongest baseline attack (LFK) and our strongest proposed attack (SGC-BIN), we plot the confusion matrices for the 15% budget in Fig. 7. Even though both LFK and SGC-BIN poison the training nodes belonging to the same two classes (class 2 and class 4), after the SGC-BIN attack the model more consistently confuses the test predictions for these two classes.

### B.8.5 EXTENDED META ATTACK

In Table 6 we evaluate the extended variant of our meta attack. Overall, we can conclude that this variant is weaker compared to the other meta variants (and our surrogate attacks) while still outperforming some of the baselines. In any case, we include these results for completeness.

### B.8.6 ATTACKING WITHOUT ACCESS TO GROUND-TRUTH TEST LABELS

In our main paper, we described a second threat model, where the ground truth labels are only available for training nodes and only the predictions are available for test nodes (no ground-truth labels). We evaluate our strongest attacks SGC-BIN and META-BIN in this setting and plot the results on Fig. 8. We observe that the performance of our attacks in this setting is close to the worst-case setting where the attacker knows the test ground-truth labels. Especially for SGC-BIN there is almost no difference between using the ground-truth labels vs. the predictions. Additionally, our attacks that use predictions also outperform the baselines that use ground-truth labels.

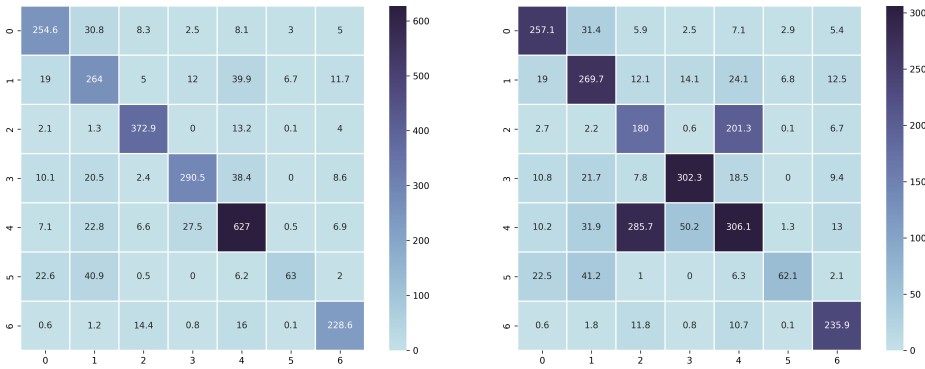

Figure 7: Confusion matrices for test set for a GCN model poisoned with the LFK (left) and SGC-BIN (right) attacks using a 15% poisoning budget on the Cora-ML dataset.

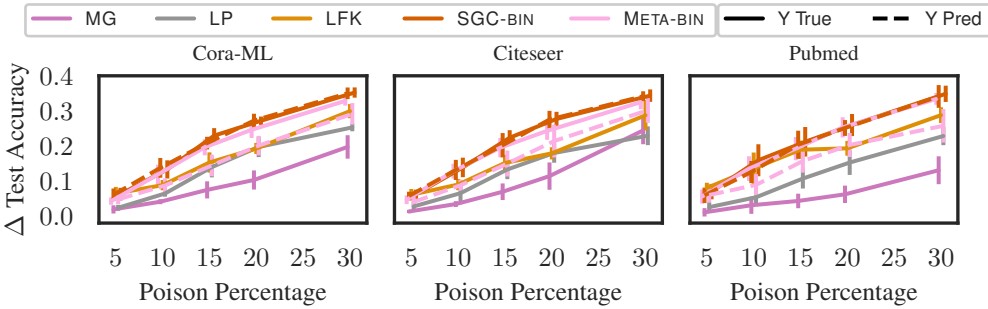

Figure 8: Evaluating our attacks under the two different threat models given access to: ground-truth test labels vs. only test predictions. Our attacks outperform the baselines even without access to ground-truth test labels.

## B.9 DATASET STATISTICS

In Table 7, we tabulate dataset statistics. We additionally include the train/val/test split statistics for the default and the proposed CV setting. Note that in the CV setting, the test accuracy is measured over all the remaining unlabeled nodes, and the train and val set have the same size.

Table 7: Dataset statistics.

| Dataset | Nodes | Features | Classes | Train/Val/Test Split | |
| --- | --- | --- | --- | --- | --- |
| | | | | **Default** | **CV** |
| **Cora-ML** | 2810 | 2879 | 7 | 140/500/1000 | 140/140/2530 |
| **Citeseer** | 2110 | 3703 | 6 | 120/500/1000 | 120/120/1870 |
| **Pubmed** | 19717 | 500 | 3 | 60/500/1000 | 60/60/19597 |
| **CoraFull** | 18800 | 500 | 70 | - | 1400/1400/16000 |
| **OGBN-ArXiv** | 169343 | 128 | 40 | 90941/29799/48603 | 90941/29799/48603 |

## B.10 TRAINING WITH A CLEAN VALIDATION-SET

As mentioned in the § 3, assuming access to a clean validation dataset is not a realistic scenario, particularly in the graph domain where we are often interested in sparsely labeled scenarios where both the training and validation sets are small and typically of the same size. Consequently, the defender can always exploit the limited clean validation set to train the model and thwart the attacker's efforts. To illustrate this further, we demonstrate that the defender can train the model using the clean validation data while maintaining performance.

In this experiment, we assume access to a small validation set with just 20 labels per class as the standard choice. This constitutes roughly 5% of Cora-ML, 6% of Citeseer, and 0.3% of Pubmed datasets. We train a GCN model for 200 epochs using this validation set and discard the potentially poisoned training set. We perform this experiment across 10 different splits. Table 8 illustrates the results, demonstrating that all label-poisoning attacks can be effectively neutralized.

## B.11 HOW LABEL POISONING INFLUENCES GRAPH HOMOPHILY

While it is a fact that label poisoning attacks can disrupt the smoothness of labels in a graph, these attacks do not provide the defender with the ability to detect the specific poisoned labels. The defender is presented with a graph that may contain tainted training labels and starts with no knowledge of the graph's homophily rate before the poisoning occurred. Moreover, real-world graphs exhibit varying levels of homophily, with examples such as OGBN-ArXiv having a 63% homophily rate, and Pubmed

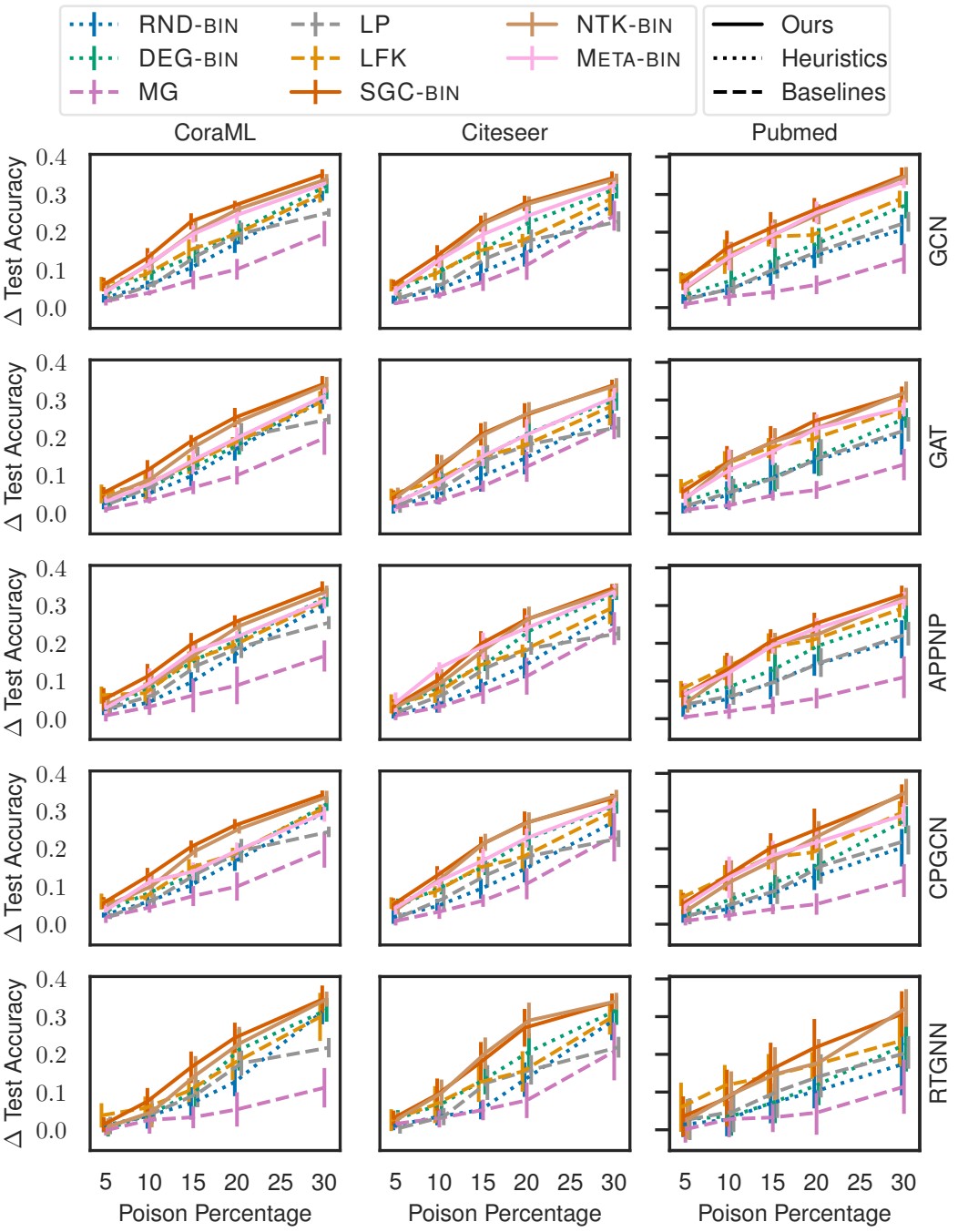

Figure 9: Our strongest attacks significantly outperform the baselines across various models and datasets.

Table 8: Test accuracy on large clean validation set as training set.

| Dataset | Test Accuracy (%) |
|---------|-------------------|
| **Cora-ML** | $85.30_{\pm 0.95}$ |
| **Citeseer** | $75.62_{\pm 0.99}$ |
| **Pubmed** | $82.40_{\pm 1.00}$ |

boasting approximately 80%. However, even if the defender knows about the graph's homophily, we demonstrate that label poisoning does not significantly change the graph's homophily. Table 9 and Table 10 illustrate how the global and local homophily rates change only minimally for different poison budgets when using the SGC-BINattack:

Table 9: Edge (global) homophily ratio.

| Dataset | 0 | 5 | 10 | 15 | 20 | 30 |
|---------|------|------|------|------|------|------|
| **Cora-ML** | 0.82 | 0.80 | 0.79 | 0.79 | 0.78 | 0.77 |
| **Citeseer** | 0.80 | 0.78 | 0.77 | 0.76 | 0.75 | 0.74 |
| **Pubmed** | 0.84 | 0.84 | 0.84 | 0.84 | 0.84 | 0.83 |

Table 10: Node (local) homophily ratio.

| Dataset | 0 | 5 | 10 | 15 | 20 | 30 |
|---------|------|------|------|------|------|------|
| **Cora-ML** | 0.82 | 0.83 | 0.83 | 0.82 | 0.81 | 0.80 |
| **Citeseer** | 0.80 | 0.80 | 0.79 | 0.78 | 0.77 | 0.76 |
| **Pubmed** | 0.84 | 0.86 | 0.86 | 0.86 | 0.86 | 0.86 |

## B.12 ATTACK PERFORMANCE UNDER THE DEFAULT (FLAWED) SETTING

We also evaluate our proposed attack under the default and flawed evaluation setting to demonstrate that our attacks outperform the baselines by an even wider margin. The reason for advocating the new evaluation protocol is to simulate the attack/defense scenario more realistically, not because it favors our attacks. Table 11 shows the results for the Cora-ML dataset, clearly demonstrating our attack's superior performance over baseline attacks.

## B.13 ATTACKS UNDER PARTIAL INFORMATION SCENARIO

We further conduct experiments involving partial information to evaluate our poisoning attack. In this context, we exclusively utilized node features and excluded the graph structure for the attacker. The attacks were executed on a graph generated from the k-nearest neighbors of the node. Table 12 illustrates our result. Our results reveal that our SGC-BIN KNN attack (with partial information, considering only node features) closely matches LFK's performance when it has access to complete graph information. Additionally, it outperforms LFK by approximately 2% on average under partial information. This highlights that our proposed attack surpasses the previous state-of-the-art method, even in scenarios with partial information.

## B.14 TIMING ANALYSIS

We compute the average time taken in seconds (over 20 different runs) for a single hyper-parameter configuration for our proposed evaluation setting and tabulate in Table 13. We can infer that tuning GNNs is relatively cheap.

Table 11: Performance comparison of proposed attacks and baselines under default evaluation setting on the Cora-ML dataset.

| Attack | 5% | 10% | 15% | 20% |
|---|---|---|---|---|
| SGC-BIN | $66.80_{\pm 2.73}$ | $60.78_{\pm 2.03}$ | $\mathbf{53.88}_{\pm 3.27}$ | $\mathbf{49.12}_{\pm 2.16}$ |
| META | $\mathbf{61.80}_{\pm 3.37}$ | $\mathbf{59.48}_{\pm 1.96}$ | $58.27_{\pm 2.05}$ | $50.30_{\pm 3.08}$ |
| LFK | $66.43_{\pm 2.29}$ | $67.33_{\pm 3.68}$ | $58.93_{\pm 2.38}$ | $55.94_{\pm 3.44}$ |
| RND | $80.03_{\pm 1.39}$ | $78.79_{\pm 2.31}$ | $76.21_{\pm 2.93}$ | $74.19_{\pm 2.84}$ |
| DEG | $79.98_{\pm 1.57}$ | $76.95_{\pm 2.25}$ | $74.28_{\pm 2.73}$ | $70.41_{\pm 2.54}$ |
| MG | $78.71_{\pm 2.54}$ | $72.08_{\pm 5.25}$ | $63.74_{\pm 6.87}$ | $58.74_{\pm 7.10}$ |

Table 12: SGC-BIN KNN and LFK KNN illustrate attacks with partial information using K-nearest neighbors (K=20) for graph construction. Meanwhile, SGC-BIN, LFK, and LP demonstrate attack performance with full information (adjacency matrix and node features).

| Cora-ML | 5% | 10% | 15% | 20% | 30% |
|---|---|---|---|---|---|
| SGC-BIN | $76.57_{\pm 1.89}$ | $69.50_{\pm 2.49}$ | $59.81_{\pm 2.23}$ | $55.55_{\pm 1.38}$ | $47.47_{\pm 1.64}$ |
| LFK | $76.38_{\pm 2.10}$ | $73.98_{\pm 1.03}$ | $67.50_{\pm 2.78}$ | $63.49_{\pm 1.26}$ | $52.80_{\pm 1.95}$ |
| LP | $80.48_{\pm 0.55}$ | $76.40_{\pm 0.58}$ | $68.83_{\pm 2.50}$ | $63.06_{\pm 2.50}$ | $57.57_{\pm 0.62}$ |
| SGC-BIN KNN | $77.99_{\pm 1.72}$ | $74.83_{\pm 2.04}$ | $67.77_{\pm 2.10}$ | $63.17_{\pm 1.82}$ | $50.43_{\pm 1.37}$ |
| LFK KNN | $77.49_{\pm 2.25}$ | $75.56_{\pm 1.54}$ | $69.47_{\pm 2.36}$ | $65.54_{\pm 1.99}$ | $53.93_{\pm 1.84}$ |

### B.15 LABEL FREQUENCY ANALYSIS

In Fig. 10, we plot the label frequency ratio for the benchmark datasets. We can observe that the frequency distribution is skewed with one or more popular classes. This reinforces our earlier point that class-equalized splits do not capture the underlying distribution of labels.

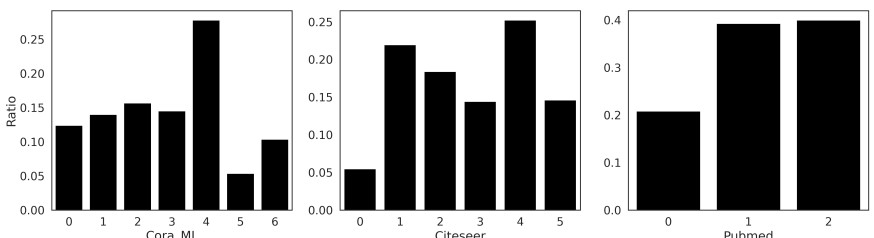

Figure 10: Label frequency ratio for benchmark datasets.

### B.16 EVALUATION AGAINST CERTIFIED DEFENSE MODEL

Label poisoning can be considered as an extreme version of feature poisoning. To explore this implication, we evaluate our strongest attack SGC-BIN and the strongest baseline attack LFK on AB-LATIONGCN, one of the recently certified defense mechanisms against feature perturbations. We tabulate our results in Table 14. We can infer that our proposed attack can inflict heavy damage on the model's performance.

Table 13: Timing analysis.

| Dataset | K-80 GPU | A100 GPU |
|---|---|---|
| **Cora-ML** | 16.3s | 10.2s |
| **Citeseer** | 18.9s | 11.8s |
| **Pubmed** | 36.4s | 12.5s |

Table 14: Efficacy of SGC-BIN vs. LFK attacks on AblationGCN, a certified defense against feature perturbations.

| | 0% | 5% | 10% | 15% | 20% | 30% |
|---|---|---|---|---|---|---|
| | | | **Cora-ML** | | | |
| **SGC-BIN** | $79.17_{\pm0.83}$ | $71.80_{\pm1.65}$ | $66.34_{\pm1.70}$ | $59.77_{\pm1.25}$ | $54.57_{\pm1.25}$ | $46.60_{\pm1.30}$ |
| **LFK** | $79.17_{\pm0.83}$ | $71.63_{\pm1.80}$ | $68.32_{\pm1.65}$ | $63.17_{\pm1.73}$ | $58.93_{\pm1.60}$ | $50.81_{\pm2.16}$ |
| | | | **Citeseer** | | | |
| **SGC-BIN** | $68.25_{\pm1.66}$ | $62.90_{\pm2.32}$ | $56.44_{\pm3.22}$ | $49.64_{\pm2.05}$ | $44.76_{\pm2.32}$ | $36.39_{\pm1.63}$ |
| **LFK** | $68.25_{\pm1.66}$ | $63.10_{\pm2.06}$ | $59.64_{\pm2.02}$ | $53.10_{\pm2.73}$ | $50.86_{\pm2.06}$ | $40.42_{\pm2.68}$ |
| | | | **PubMed** | | | |
| **SGC-BIN** | $78.07_{\pm1.17}$ | $67.02_{\pm1.81}$ | $61.38_{\pm2.64}$ | $56.40_{\pm2.81}$ | $53.73_{\pm1.99}$ | $47.77_{\pm2.01}$ |
| **LFK** | $78.07_{\pm1.17}$ | $67.36_{\pm1.43}$ | $61.80_{\pm4.73}$ | $57.07_{\pm1.92}$ | $55.87_{\pm2.74}$ | $50.35_{\pm2.30}$ |

## B.17 ABLATION STUDY ON PITFALLS

In this section, we expand upon the experiments conducted in the main paper by including additional datasets and models. The majority of our observations in Fig. 11, Fig. 12, Fig. 13, Fig. 14, Fig. 15 align with the findings reported in the main paper. Specifically, for the Citeseer and Pubmed datasets, we notice that, in certain budget scenarios, our proposed CV splits appear to be advantageous to the attacker when compared to the default configuration for the LFK attack.

## B.18 CONNECTIONS BETWEEN PITFALLS AND ATTACKS

Our main goal is to establish a principled evaluation protocol to ensure that research on label poisoning (for ourselves and others) is on a solid ground. We see two equally important parts to this goal:

- Evaluation setup that simulates the full potential of a defender as realistically as possible.
- Simple yet effective baseline attacks to serve as reference point for the future.

These two parts are addressed by the (remedies of the) pitfalls and the attacks. On one hand, you can definitely see them as two disjoint contributions. On the other hand, having a fair evaluation in mind, implicitly guided our design of the attacks. For example, pitfalls P3 and P6 can be seen as a form of overfitting to the splits and default h-params respectively. Therefore, when selecting h-params of the attacks (e.g. the value of $\lambda$ for LSA attacks and the learning rate for meta attacks) we choose h-params that minimize the test accuracy on average across splits, preventing overfitting to a single split. Moreover, to remedy P6 we poison $\mathcal{V}_l$ which implicitly impacts the bi-level optimization. Moreover, while fixing the identified pitfalls leads to a fairer evaluation it also decreases the effectiveness of previous attacks. This provided additional motivation to introduce the new baseline attacks. Under the previous evaluation setting, we observe that META attacks significantly outperform other attacks (see § B.12). However, in the new fairer setting, META attacks don't generalize as effectively as linear attacks. This highlights the need for a careful evaluation.

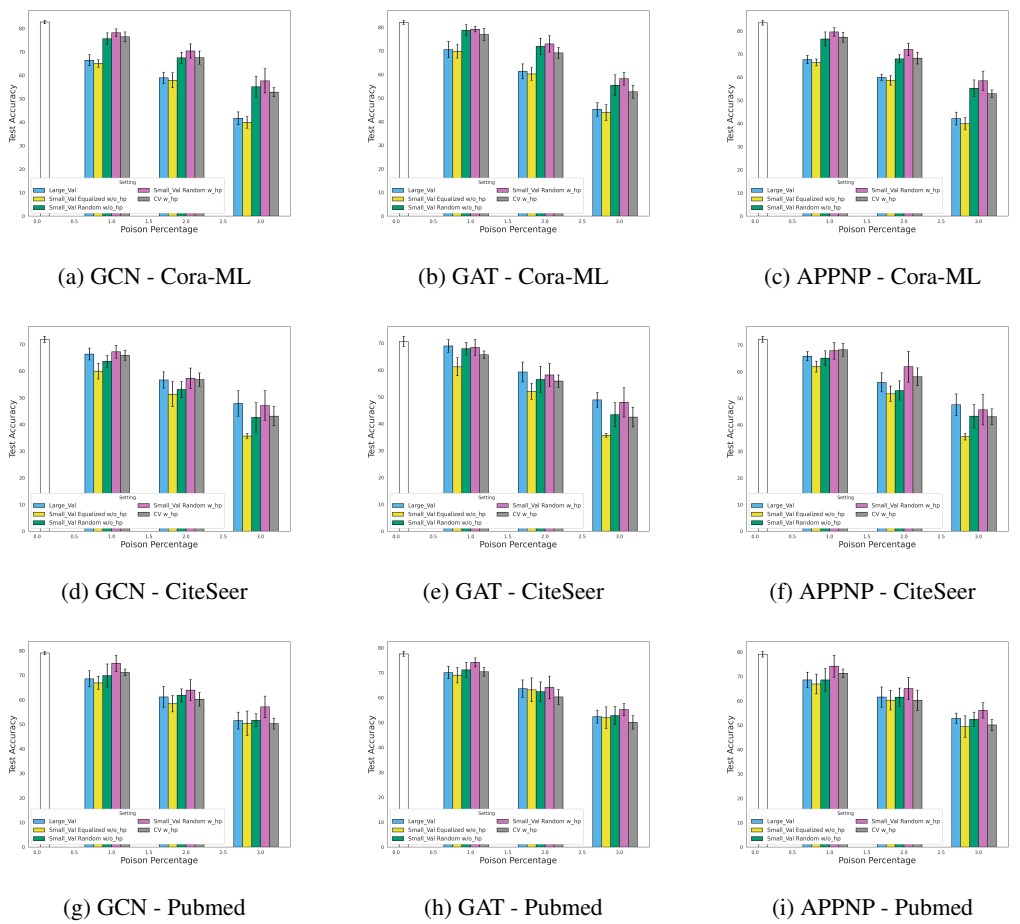

Figure 11: Analyzing the effect of each evaluation setting on the performance of the LFKattack across three models and three datasets.

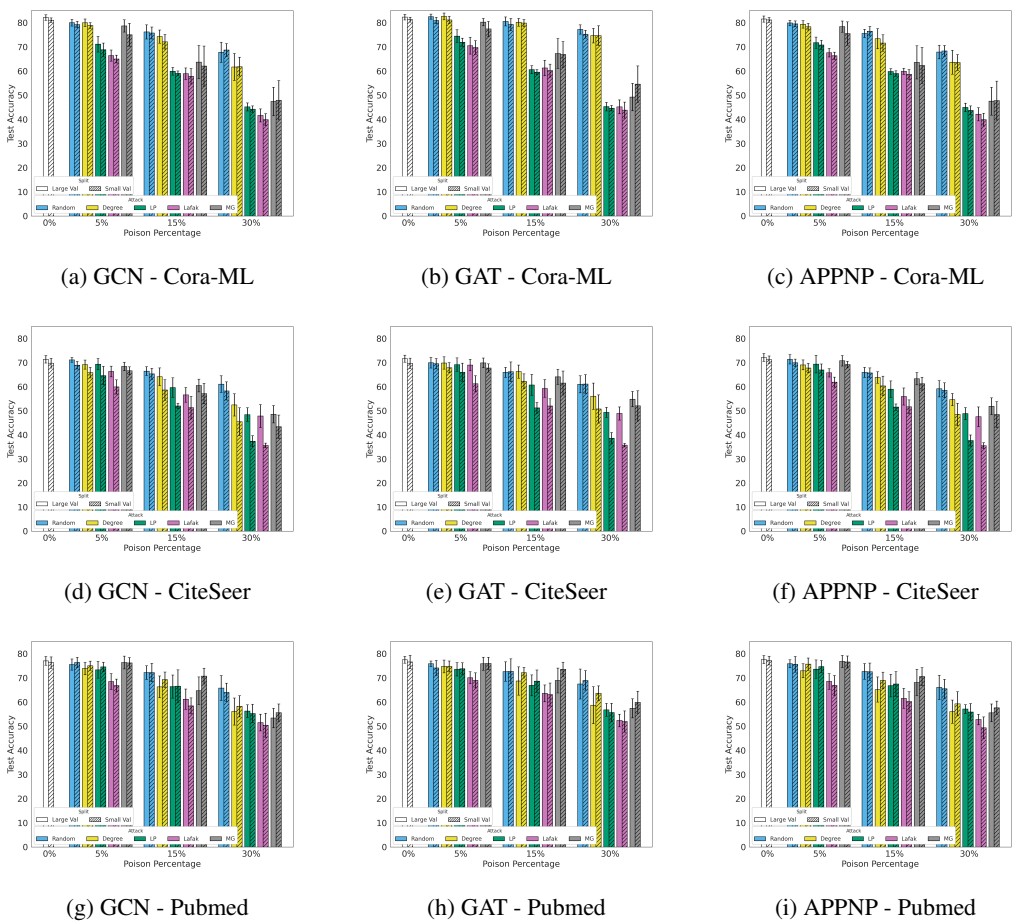

Figure 12: Effect of validation set size on the performance of poisoning attacks across three models and three datasets.

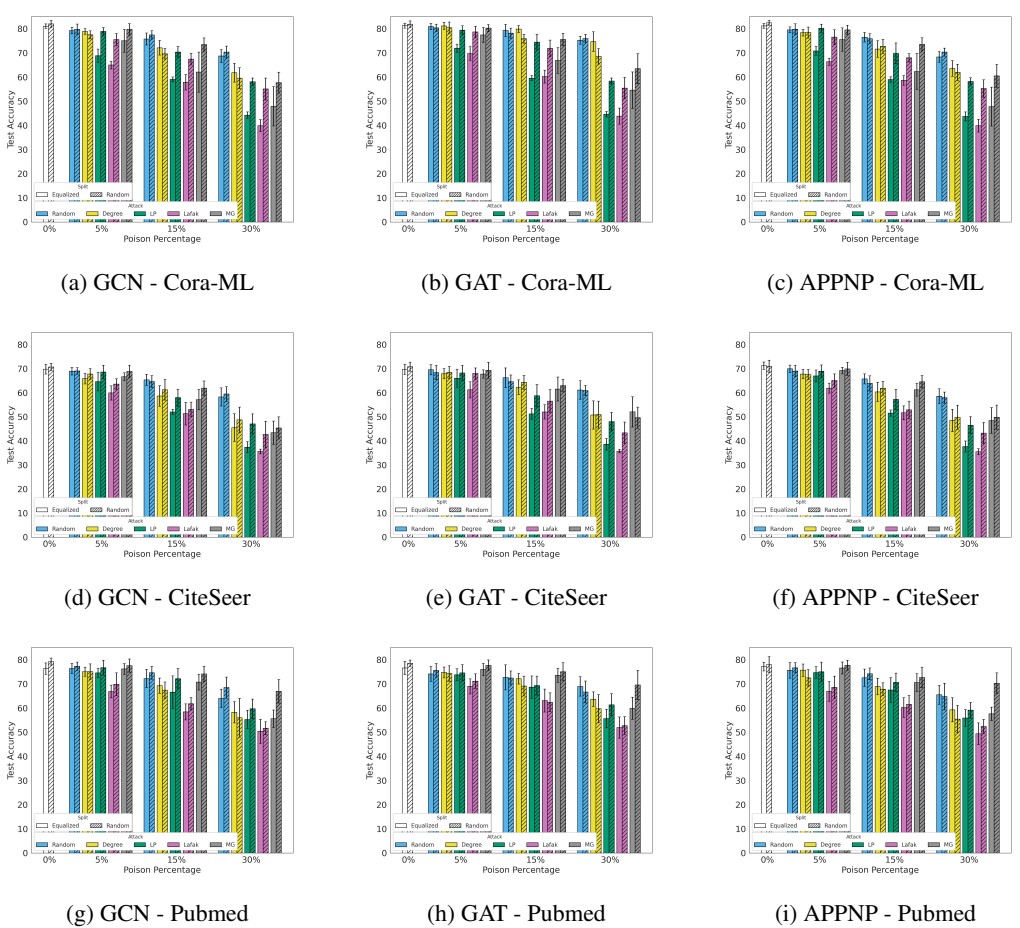

Figure 13: Effect of data split on the performance of poisoning attacks across three models and three datasets.

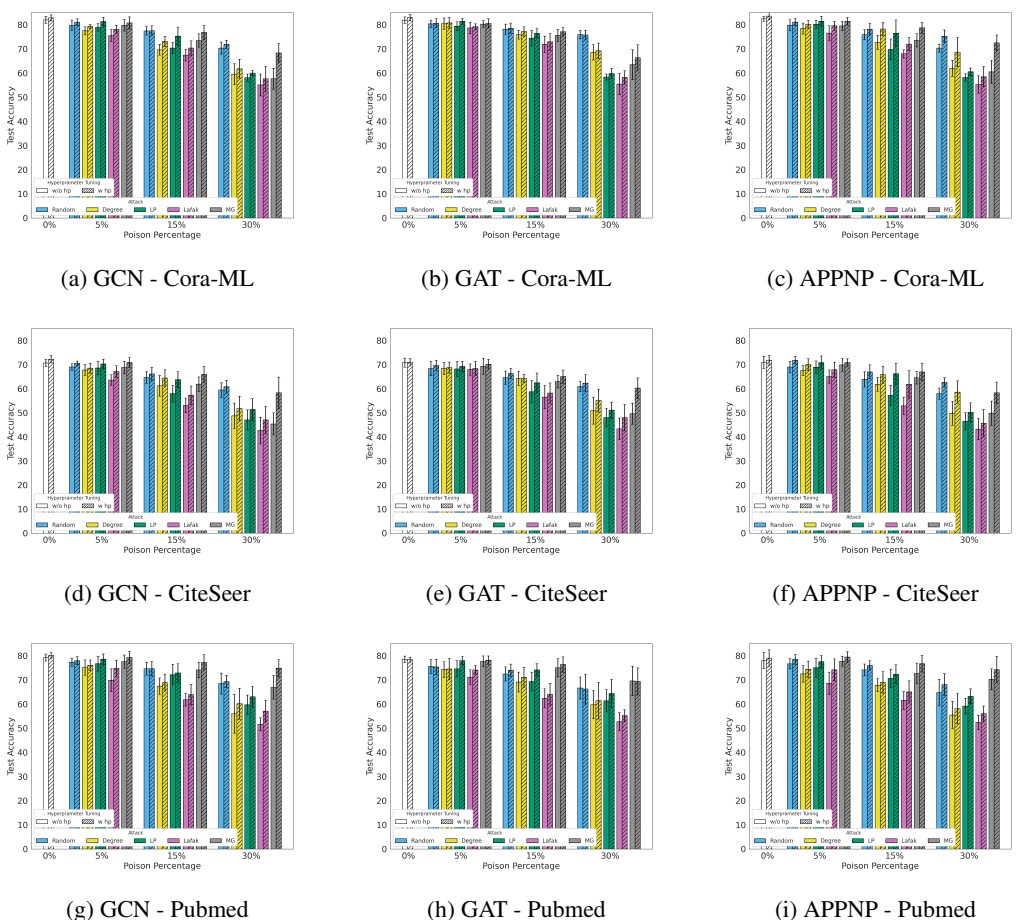

(a) GCN - Cora-ML     (b) GAT - Cora-ML     (c) APPNP - Cora-ML

(d) GCN - CiteSeer     (e) GAT - CiteSeer     (f) APPNP - CiteSeer

(g) GCN - Pubmed     (h) GAT - Pubmed     (i) APPNP - Pubmed

Figure 14: Analyzing the effect of proper hyper-parameters tuning on the performance of poisoning attacks across three models and three datasets.

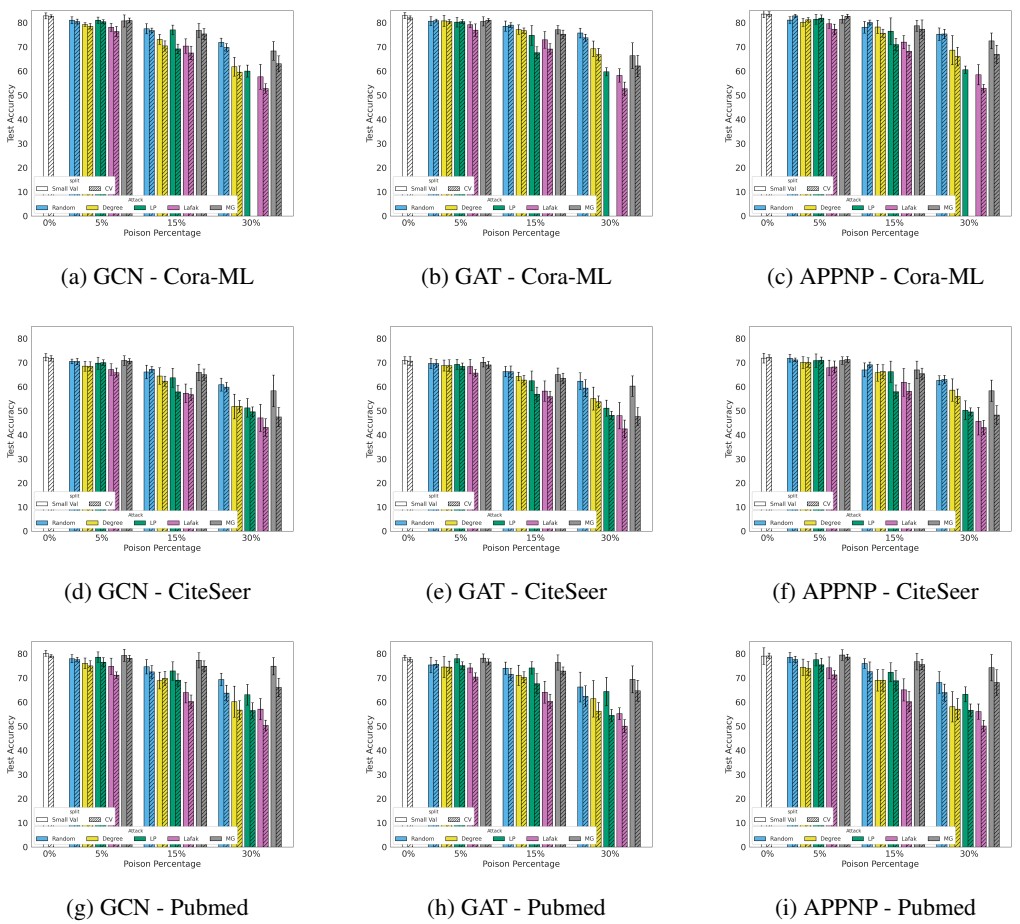

Figure 15: Analyzing the effect of proper cross-validation on the evaluated performance of poisoning attacks across three models and three datasets.

