# OpenReview forum: "Rethinking Label Poisoning for GNNs: Pitfalls and Attacks"
_ICLR.cc/2024/Conference — ICLR 2024 poster_

### Official Review · Reviewer_Cen8 · 2023-10-29

**Soundness:** 2 fair
**Presentation:** 2 fair
**Contribution:** 2 fair
**Rating:** 6
**Confidence:** 5

**Summary:**

This paper evaluates the current state of research on label poisoning in Graph Neural Networks (GNNs). The authors point out that while noisy labels in machine learning have been extensively studied, the same cannot be said for graph-based data, especially in the context of adversarial label perturbations. The paper identifies evaluation pitfalls in existing literature, questioning the conclusions drawn about the robustness of GNNs against label poisoning. To address this, the authors delve into various attack strategies, explore the intricacies of loss functions, and aim to provide a more accurate understanding of GNN vulnerabilities.

**Strengths:**

1. Formalisation: This paper provides a formalisation for the research questions.
2. Focus on Nuances: A lot of part of this paper focuses on discriminating the nuances of existing methods, which contributes to building trustworthy GNNs from an accountability view. The authors don't just critique existing literature but also provide a summarisation of pitfalls in existing works.

**Weaknesses:**

1. Unconvincing arguments, presentation ambiguity and self-contradiction: (1) Lot of arguments in this paper are not convincing. For example, it says “The first pitfall arises since all previous attacks evaluate on data splits with validation set size much larger than that of the training set (e.g. 500 validation vs 140 train nodes on Cora-ML in the default setting).” Actually, this kind of data partitioning is practical in graph learning as model developers focus on training and validating datasets in the training phase. It is hard to say this is a pitfall. (2) A lot of the statements are ambiguous. For example, it says “We observe that fine-tuning with only 20 configurations can significantly deteriorate the attack performance, as evidenced by an increase in test accuracy.” The configurations are not explained clearly here. (3) Self-contradictive statement. In the beginning, it says that one drawback of related works is they focus on the binary task, however, they can be generalised to multiclass tasks, as shown in the evaluation part of this paper.
2. Limited novelty. Although the research problem has been formulated, the proposed solutions have limited contribution. First, the method in section 4.1, only focuses on the liner GCN model, which ignores the non-linear property of most GNN models. The second method introduces a learning-based method to select nodes for flipping, however, the effectiveness will be limited by the expression ability of the surrogate model in the inner optimisation. Considering these limitations, existing methods can be adapted to the current setting, and unhighlighted research challenges in this paper, the novelty of the proposed methods is weak.
3. Objective Presentations. Broad claims, especially critiques of an entire domain of literature, require comprehensive evidence and careful presentation to avoid potential overgeneralization. For example, the statement that the "entire literature on label poisoning for GNNs is plagued by serious evaluation pitfalls" is a strong assertion. From the segments I reviewed, there wasn't exhaustive and convincing evidence presented to substantiate such a broad critique. While the paper might delve deeper into this in sections I haven't extracted, from the available content, this claim appears to lack detailed supporting examples.

A research paper's strength lies not just in its findings but also in the clarity and robustness of its evaluation methods. When critiquing other works for evaluation pitfalls, it becomes crucial for the paper to transparently outline its own methodology limitations. Considering these weaknesses in this paper, the authors are suggested to polish this paper carefully.

**Questions:**

Refer to weakness part.

---

> ### Author Response · Authors · 2023-11-16
> **Response (part-1)**
>
> We thank the reviewer for their valuable feedback. We address the raised questions below.
>
> ---
> **W1-1.** *Lot of arguments in this paper are not convincing. For example, it says “The first pitfall arises since all previous attacks evaluate on data splits with validation set size much larger than that of the training set (e.g. 500 validation vs 140 train nodes on Cora-ML in the default setting).” Actually, this kind of data partitioning is practical in graph learning as model developers focus on training and validating datasets in the training phase. It is hard to say this is a pitfall.*
>
> **Response W1-1.** We strongly disagree with this comment. Unlike other domains (e.g. Computer Vision) in GNNs we are often interested in the sparsely labeled scenario where the number of labeled nodes is relatively small $\mathcal{V}_l \ll \mathcal{V}$. This is the standard in the literature.
>
> We advocate using validation set that is at most as large as the train set. The pitfall of using large validation set has been extensively studied for semi-supervised learning. The highly influential paper from Oliver et al. (2018) [1] offers an extensive discussion on this pitfall for general semi-supervised learning (see their P.6). To briefly summarize their points, in real-world applications the larger validation set would instead be used as the train set. Another issue is that any objective (e.g. accuracy) used for hyperparameter tuning would be significantly noisier across runs when using a realistically small validation set which disproportionately affects method that are sensitive to h-params.
>
> It would benefit the discussion if the reviewer can point us to other arguments that they find unconvincing.
>
> [1] Realistic Evaluation of Deep Semi-Supervised Learning Algorithms. Oliver et al., NeurIPS 2018.
>
> ---
> ---
> **W1-2.** *A lot of the statements are ambiguous. For example, it says “We observe that fine-tuning with only 20 configurations can significantly deteriorate the attack performance, as evidenced by an increase in test accuracy.” The configurations are not explained clearly here.*
>
> **Response 1-2.** In the introductory paragraph of our Appendix we provide comprehensive details on the hyper-parameter configurations we use during both attack (paragraph 2) and defense (paragraph 3) phases. We added a forward pointer for clarity in the updated version. Additionally, we share our code which does hyper-parameter tuning and provides additional implementation level details.
>
> It would benefit the discussion, if the reviewer can point us to other statements that are ambiguous.
>
> ---
> ---
>
> **W1-3.** *Self-contradictive statement. In the beginning, it says that one drawback of related works is they focus on the binary task, however, they can be generalised to multiclass tasks, as shown in the evaluation part of this paper.*
>
> **Response W1-3.**  We mentioned in our paper that the previous attacks were primarily introduced for binary classes. Then, for a fair comparison, we introduced the multi-class extension of these attacks. We therefore fail to see the contradiction that the reviewer is pointing out to.
>
> ---
> ---
>
> **W2.** *Limited novelty. Although the research problem has been formulated, the proposed solutions have limited contribution. First, the method in section 4.1, only focuses on the liner GCN model, which ignores the non-linear property of most GNN models. The second method introduces a learning-based method to select nodes for flipping, however, the effectiveness will be limited by the expression ability of the surrogate model in the inner optimisation. ...*
>
> **Response W2.** We provide clarification on our proposed approaches. It is important to note that our simplest linear surrogate attacks, despite not retaining the non-linearity property, are very effective on other non-linear GNN models like GCN, RTGNN, GAT etc. This highlights the transferability of our attacks. In addition to this, we also introduce NTK attacks that take into account the non-linearity of GCNs. This is expected since linear GNNs perform competitively to non-linear GNN on several homophilous datasets [1, 2].
>
> For the LSA family of attacks, we prove that the MILP can be efficiently solved (see Proposition 1 for a closed-form solution and Proposition 2 for the proof that the LP is integral). These results are novel and show that the globally optimal MILP solution can be efficiently computed.
>  We also prove that random binary attacks are stronger than multi-class variants (Proposition 3).
>
> The Meta attacks that we introduce do not use any surrogate models in the inner-optimization. Our adaptive meta attacks uses the exact same model and not a surrogate model in the inner optimization. For additional implementation-level insights, please refer to our code.
>
> [1] Simplifying Graph Convolutional Networks. Wu et al., ICML 2019
>
> [2] Understanding Non-linearity in Graph Neural Networks from the Bayesian-Inference Perspective. Wei et al., NeurIPS 2022

---

> > ### Author Response · Authors · 2023-11-16
> > **Response (part-2)**
> >
> > **W3.** *Objective Presentations. Broad claims, especially critiques of an entire domain of literature, require comprehensive evidence and careful presentation to avoid potential overgeneralization. For example, the statement that the "entire literature on label poisoning for GNNs is plagued by serious evaluation pitfalls" is a strong assertion. From the segments I reviewed, there wasn't exhaustive and convincing evidence presented to substantiate such a broad critique. While the paper might delve deeper into this in sections I haven't extracted, from the available content, this claim appears to lack detailed supporting examples.*
> >
> > **Response W3.** We thank the reviewer for their feedback. While our statements are true they might be considered misleading since there are only a handful of papers on adversarial label poisoning attacks for GNNs. We appropriately softened our statements.
> >
> > ---
> > ---
> >
> > **W4.** *"…When critiquing other works for evaluation pitfalls, it becomes crucial for the paper to transparently outline its own methodology limitations."*
> >
> > **Response W4.** We already highlighted in the paragraph on Ablations and Limitations (refer to end of Section 5) the limitations of our work. Pasting below the two main limitations of our work for ease of reference.
> >
> > ```
> > While it may not always be easy to implement our threat models in practice,
> > we still gain significant insights into the inherent resilience of the models. Our theoretical insights
> > (Proposition 3) apply only to random flips, and extending this to the worst-case is left for future work
> > ```
> >
> > We are happy to incorporate any additional limitations that the reviewer might have identified.

---

> > ### Comment · Reviewer_Cen8 · 2023-11-17
> > **For the Response to W2**
> >
> > Actually, I suppose that this discussion is about the attack setting of this paper. According to the knowledge of attackers, existing attacks on a model can be categorised into white-box, grey-box, and black-box settings.
> > In the black-box setting, attackers have no knowledge about the target model; while in the white the white-box setting, attackers have knowledge about the target model, e.g., architecture, and parameters.
> >
> > In the context of this paper, Label Poisoning for GNNs, attackers generally do not have knowledge of the target model, as they cannot expect the model developer to use the specific architecture (e.g., SGC) as how to choose the model architecture depending on model developer but not attacker. In this case, the attacker usually uses a surrogate model (e.g., SGC) to mimic the learning of the target model, which can be used to generate a poisoned dataset. However, the generated poisoned data depends on the surrogate model, which is potentially ineffective when current graph structure learning methods are introduced in the training of the target GNN model. This is why I proposed the limitation of this paper.
> >
> > Given this background, the response of the authors seems confusing and cannot address the proposed weakness. It supports the above pipeline (i.e., black-box attack) as it says "It is important to note that our simplest linear surrogate attacks, despite not retaining the non-linearity property, are very effective on other non-linear GNN models like GCN, RTGNN, GAT etc. This highlights the transferability of our attacks". However, it also shows that "our adaptive meta attacks use the exact same model and not a surrogate model in the inner optimization.", which supports attackers knowing the architecture of the target model in advance.

---

> > > ### Author Response · Authors · 2023-11-17
> > > **Response**
> > >
> > > Thank you for this observation. Indeed, linear and meta attacks can be seen as addressing two different threat models, black-box and white-box respectively. In our view, including this perspective only strengthens the paper, since each of the baseline families of attacks covers a different threat model. Why is this a limitation?
> > >
> > > Note, we also already covered another dimension of the threat model (access to labels and/or A) as described in paragraph 3 of section 2 which can be seen as a grey-box threat model.
> > >
> > > Moreover, our experiments show that linear attacks are indeed effective and are not "potentially ineffective".

---

> > > > ### Comment · Reviewer_Cen8 · 2023-11-21
> > > >
> > > > Dear authors,
> > > >
> > > > Thanks for your efforts in clarifying your paper. Hope my comments help to improve the manuscript. Your answers make sense to me and I have raised the score.

---

### Official Review · Reviewer_FFMY · 2023-11-05

**Soundness:** 3 good
**Presentation:** 4 excellent
**Contribution:** 4 excellent
**Rating:** 8
**Confidence:** 3

**Summary:**

This paper investigates label poisoning attacks against graph neural networks (GNNs). The paper first argues that major flaws exist in the evaluation of existing attacks against GNNs that needs to be resolved. These issues, such as the distribution of training/validation nodes and hyper-parameter selection, are related to faithfully considering the capabilities of defenders against such label poisoning attacks. The paper provides a critical evaluation of these pitfalls and demonstrate that they can have a significant effect on the performance of existing attacks. In the second part of the paper, two novel label poisoning attacks are introduced. The first one, called linear surrogate attack, uses a linear model as a surrogate for the models that one aims to attack. Then, a novel mixed-integer linear program is proposed to find the optimal solution for this attack. The second attack, dubbed meta attack, is motivated through the use of meta-learning for solving the bi-level optimization problem used for generating the label poisoning attack. Finally, the paper investigates the reasons behind superior performance of label poisoning attacks that only consider two classes to perform their flipping. Empirical and theoretical results for all the contributions are provided. Interestingly, the proposed linear surrogate attacks outperform all existing label poisoning attack baselines on both vanilla and defended GNNs.

**Strengths:**

- This paper pins down vital cracks in evaluation of label poisoning attacks and show that they can play a crucial role on the model performance. The authors present these pitfalls and argue around the unrealistic nature of each one. These arguments are supported by extensive experimental results, validating the authors' claims.

- The presented linear surrogate and meta attacks are novel and effective. The paper presents these two new methods and introduces new theory to support such attacks. Experimental results also demonstrate that these two attacks can outperform state-of-the-art baselines by a large margin.

- The most important part of this submission, in my view, is its writing and presentation. Instead of presuming that the reader is familiar with this topic, the authors present their ideas in detail and explain the intuition behind each step, making it extremely easy for the reader to navigate through the paper. Given the abundance of contributions in the paper, it could have been easy to drown the reader with a flow of information, but the authors present all these ideas in a coherent manner, making the paper a pleasure to read.

**Weaknesses:**

- Perhaps the most problematic issue with the current submission for me is the lack of direct relationship between the proposed methods and the discussions of the first half of the paper. It would have been nicer if the authors could make the connection of these two parts more clear. In other words, was the design of the proposed attacks in any form motivated by the flaws in evaluating label poisoning attacks in GNNs, or these two parts of the paper shall be seen as two disjoint contributions?

- Besides, providing some explanation/intuition about the first pitfall of baseline evaluation would be nice. At the moment, the paper just states that existing methods use considerably larger validation sets than training ones, but it doesn't go into the details of why this is a pitfall. Elaborating on the during the second paragraph of Section 3 is much appreciated.

**Questions:**

Besides the above-mentioned questions, here are some additional questions/suggestions:

- Could you please explain how the final poisoned label is computed via $\hat{\boldsymbol{Y}}\_{l} = \mathrm{diag}(\boldsymbol{b}) \odot \boldsymbol{H} + \mathrm{diag}(\mathbf{1}\_{L} - \boldsymbol{b}) \odot \boldsymbol{Y}\_{l}$?

- Using larger legends and axis labels for the plots is highly encouraged.

---

> ### Author Response · Authors · 2023-11-16
>
> We thank the reviewer for their valuable feedback. We address the raised questions below.
>
> ---------
> **W1.** *Perhaps the most problematic issue with the current submission for me is the lack of direct relationship between the proposed methods and the discussions of the first half of the paper. It would have been nicer if the authors could make the connection of these two parts more clear. In other words, was the design of the proposed attacks in any form motivated by the flaws in evaluating label poisoning attacks in GNNs, or these two parts of the paper shall be seen as two disjoint contributions?*
>
> **Response R1.** Since a similar comment was brought up by multiple reviewers we address it in a joint meta response (see the top comment and the newly added section A.18).
>
> -------------
> -------------
>
> **W2.** *Besides, providing some explanation/intuition about the first pitfall of baseline evaluation would be nice. At the moment, the paper just states that existing methods use considerably larger validation sets than training ones, but it doesn't go into the details of why this is a pitfall. Elaborating on the during the second paragraph of Section 3 is much appreciated.*
>
> **Response W2.** We elaborated this point in the updated version. The highly influential paper from Oliver et al. (2018) [1] offers an extensive discussion on this pitfall for general semi-supervised learning (see their P.6). To briefly summarize their points, in real-world applications the larger validation set would instead be used as the train set. Another issue is that any objective (e.g. accuracy) used for hyperparameter tuning would be significantly noisier across runs when using a realistically small validation set which disproportionately affects method that are sensitive to h-params.
>
> [1] Realistic Evaluation of Deep Semi-Supervised Learning Algorithms. Oliver et al., NeurIPS 2018.
>
> -------------
> -------------
>
> **Q1. Could you please explain how the final poisoned label is computed via...**
>
> **Response.** Instead of Hadamard product, we need a matrix product. Thank you for noticing this typo.
>
> Correct version: $\widehat{\mathbf{Y}}_l =  \mathrm{diag}(\mathbf{b}) \mathbf{H} + \mathrm{diag} (\mathbf{1}_L - \mathbf{b})  \mathbf{Y}_l $.
>
> We also updated the paper to reflect this.

---

### Official Review · Reviewer_82Ci · 2023-11-06

**Soundness:** 2 fair
**Presentation:** 3 good
**Contribution:** 2 fair
**Rating:** 5
**Confidence:** 3

**Summary:**

The paper first analyze the exiting problems in the current evaluation on the label-poisoning attack in the graph data. Through the analysis, the paper finds that the current evaluation has validation/test data unbalanced problem and is only tested in a single hyperparameter setting. The evaluation also keeps the labels in the validation/test data is clean. Then the paper proposes a new attack that formalize the label poisoning attack problem into a mixed-integer linear program. The paper also proposes to try  an orthogonal approach where it directly optimize the poisoned label through gradient descent and use the gumbel-softmax loss to enforce the discrete property. Extensive experiments have been conducted to verify the effectiveness of the proposed method.

**Strengths:**

1. The paper is well-written and not hard to follow.
2. The analysis of current problem in the evaluation does help the future study in the label poisoning community.

**Weaknesses:**

1. The paper is not well-organized. The analysis on current evaluation doesn't help to understand why the new method is proposed. The paper does want to include a lot of discussion on different parts of the label-poisoning attack. However, I find they are not well-connected and I fail to find a coherent logic flow in the paper.
2. Since I am not from this community, the proposed method seems a very natural formulation to conduct the label flip idea. Therefore, I am not sure whether the paper has enough novelty. Also, the formulation is also heavily based on the previous study and the proposed method is like a multi-label extension to me.
3. Some pitfalls proposed are weird to me. I am not sure why the split between training, validation and test should be equal. As a semi-supervised learning task, it is usually to let them differ in my opinion and I am also not sure why the validation set should be poisoned as well since the validation set should be used to select model and do cross-validation.

**Questions:**

Please refer to the weaknesses.

---

> ### Author Response · Authors · 2023-11-16
>
> We thank the reviewer for their valuable feedback. We address the raised questions below.
>
> -------------------
>
> **W1.** *The paper is not well-organized. The analysis on current evaluation doesn't help to understand why the new method is proposed. The paper does want to include a lot of discussion on different parts of the label-poisoning attack. However, I find they are not well-connected and I fail to find a coherent logic flow in the paper.*
>
> **Response W1.** Since a similar comment was brought up by multiple reviewers we address it in a joint meta response (see the top comment and the newly added section A.18).
>
> ------------------------
> ------------------------
>
> **W2.** *Since I am not from this community, the proposed method seems a very natural formulation to conduct the label flip idea. Therefore, I am not sure whether the paper has enough novelty. Also, the formulation is also heavily based on the previous study and the proposed method is like a multi-label extension to me.*
>
> **Response W2.** You are correct our attacks are essentially a multi-label extension in terms of formulation. However, our novelty lies in the way we approach the optimization.
>
> **Optimal Linear Family of Attacks:** In contrast to previous approaches that rely on a greedy solution for identifying poisoned labels, our solution is optimal. Specifically, for the LSA family of attacks, we demonstrate the efficiency of solving the Mixed-Integer Linear Program (MILP) with a closed-form solution (refer to Proposition 1) and prove the integrality of the Linear Program (LP) in Proposition 2. These findings are novel and highlight the computational efficiency of obtaining the globally optimal MILP solution.
>
> **Adaptive Meta attacks:** At a high level, all meta attacks follow a similar routine -- unrolling the inner level optimization for K steps for each outer loop iteration. While various meta gradient-based attacks exist, what distinguishes them is the nuanced setup of the poisoning process. To the best of our knowledge, our work is the first to introduce a family of meta-attacks for label poisoning for GNNs. Our formulation utilizes soft-top-k followed by k-subset-selection along with the proposed Gumbel-softmax loss.
>
> Furthermore, we prove that random binary attacks surpass their multi-class counterparts (Proposition 3). Additionally, we introduce Neural Tangent Kernel (NTK) attacks.
>
> ------------------------
> ------------------------
>
> **W3-1.** *Some pitfalls proposed are weird to me. I am not sure why the split between training, validation and test should be equal. As a semi-supervised learning task, it is usually to let them differ in my opinion…*
>
> **Response W3.** Only the training and validation sets are equal and the remaining nodes correspond to the test set. We advocate using validation set that is at most as large as the train set. The pitfall of using large validation set has been extensively studied for semi-supervised learning. The highly influential paper from Oliver et al. (2018) [1] offers an extensive discussion on this pitfall for general semi-supervised learning (see their P.6). Briefly, one of their critical points is that in real-world applications the larger validation set would instead be used as the train set.
>
> [1] Realistic Evaluation of Deep Semi-Supervised Learning Algorithms. Oliver et al., NeurIPS 2018.
>
> ------------------------
> ------------------------
>
> **W3-2.** *"… and I am also not sure why the validation set should be poisoned as well since the validation set should be used to select model and do cross-validation."*
>
> **Response W3-2.** The basis for poisoning attacks is that we do now know which labels are potentially poisoned. If the defender does have access to a small, clean-label validation set, they could simply ignore the training set and train for a fixed number of epochs on this secure clean validation set. This would render all label-poisoning attacks completely ineffective. We demonstrate this with the experiment in section A.10 in the appendix.
>
> This is more acute for sparsely-labeled node classification. Unlike other domains (e.g. computer vision) in GNNs we are often interested in the sparsely labeled scenario where the number of labeled nodes is relatively small $\mathcal{V}_l \ll \mathcal{V}$. This is the standard in the literature. Some methods, use all labels for training, while others split $\mathcal{V}_l$ into train/val sets and use the latter for early stopping/fine-tuning. Our attacks remain agnostic to this.
>
> ----------------------

---

### Official Review · Reviewer_JRv5 · 2023-11-08

**Soundness:** 4 excellent
**Presentation:** 3 good
**Contribution:** 3 good
**Rating:** 8
**Confidence:** 3

**Summary:**

The paper focus on adversarial attacks in the context of graph learning algorithms. In particular it considers label poisoning (a specific case of data poisoning) for GNN algorithms. This work highlights the vulnerability of models to these attacks and the methodological problems of previous studies in evaluating this vulnerability. It also presents new attacks and provides some new insights on adversarial attacks “overfitting”.

**Strengths:**

* The paper is well written and well structured
* Pushing for higher standards in adversarial attacks is important for the research community
* The discussion on binary attacks overfitting is interesting (and would worth being investigated more)

**Weaknesses:**

* t is not super clear if the newly proposed approach are using insights coming from the different pitfalls. Especially the HP tuning one.
* Paragraph 3 mentions the pitfalls of undefended models but defers it to paragraph  ; as a result we cannot get a complete picture of the impact on the performance (and we expect this pitfall to have a large impact as well)

**Questions:**

* In Eq. (1) could we replace the argmax on L by an argmax on another metric (error rate) without loss of generality ? What would be the impact of having a more general formulation here ? (aka adversary trying to maximize error rate and not necessarily maximize loss ; would it change the design typically of the Meta attack ?)
* As mentioned above, paragraph 3 do not adress the defense-awareness pitfall ; would it be feasible to have a Figure like Fig 2 that takes this pitfall into account ? (if you have the experiments ready)
* I would suggest to add a number (P1, ..., P6) to refer to each of the pitfalls, which would make the paper easier to read (+ it would also improve the readability of some figures)
* You are introducing two new attacks. Did you leverage some insights coming from the 6 pitfalls to define these attacks ?

---

> ### Author Response · Authors · 2023-11-16
>
> We thank the reviewer for their valuable feedback. We address the raised questions below.
>
> -----------
> **W1.** *It is not super clear if the newly proposed approach are using insights coming from the different pitfalls. Especially the HP tuning one.*
>
> **Q4.** *You are introducing two new attacks. Did you leverage some insights coming from the 6 pitfalls to define these attacks ?*
>
> **Response W1/Q4.** Since a similar comment was brought up by multiple reviewers we address it in a joint meta response (see the top comment and the newly added section A.18).
>
> -----------------
> -----------------
>
> **W2.** *Paragraph 3 mentions the pitfalls of undefended models but defers it to paragraph ; as a result we cannot get a complete picture of the impact on the performance (and we expect this pitfall to have a large impact as well)*
>
> **Response W2.** The third pitfall we've identified stems from the fact that prior studies evaluated their attacks on undefended models, such as vanilla GCN and GAT, without considering defense models like CPGCN and RTGNN designed to counter label poisoning attacks.  In our evaluation, we include these defenses and additionally AblationGCN -- a certified defense against feature attacks which can be viewed as an extreme version of label poisoning attack (refer to Fig. 4).
>
> -----------------
> -----------------
>
> **Q1.** *In Eq. (1) could we replace the argmax on L by an argmax on another metric (error rate) without loss of generality ? What would be the impact of having a more general formulation here ? (aka adversary trying to maximize error rate and not necessarily maximize loss ; would it change the design typically of the Meta attack ?)*
>
> **Response Q1.** We appreciate your observation. As suggested, we acknowledge that the outer loss (L) is flexible and can be replaced with various metrics.  For example, in Sec 4.2 we introduce an approximate differentiable variant of the 0-1 loss in our meta-attack which we term as the Gumbel-softmax loss. Ablative experiments in Section A.8.2 demonstrate the superior performance of the Gumbel-softmax loss compared to the cross-entropy loss.
>
> -----------------
> -----------------
>
> **Q2.** *As mentioned above, paragraph 3 do not address the defense-awareness pitfall ; would it be feasible to have a Figure like Fig 2 that takes this pitfall into account ? (if you have the experiments ready)*
>
> **Response Q2.** In the updated version of the paper we updated Fig. 2 where now the second subplot includes a defense (CPGCN) thus showing how the second pitfall can be addressed.
>
> -----------------
> -----------------
>
> **Q3.** *I would suggest to add a number (P1, ..., P6) to refer to each of the pitfalls, which would make the paper easier to read (+ it would also improve the readability of some figures)*
>
> **Response Q3.** Thank you for the great suggestion. In the updated paper we marked each pitfall with P1 – P6 for clarity and also updated Fig. 2 using these identifiers.
>
> -------------------

---

### Author Response · Authors · 2023-11-16
**Meta Response**

We thank the reviewers for the valuable feedback. Three of the reviewers brought up very good question about the connection between the pitfalls and the attacks. We address this here.

Our main goal is to establish a principled evaluation protocol to ensure that research on label poisoning (for ourselves and others) is on a solid ground. We see two equally important parts to this goal:
- Evaluation setup that simulates the full potential of a defender as realistically as possible
- Simple yet effective baseline attacks to serve as reference point for the future

These two parts are addressed by the (remedies of the) pitfalls and the attacks. On one hand, you can definitely see them as two disjoint contributions.

On the other hand, having a fair evaluation in mind, implicitly guided our design of the attacks. For example, pitfalls P3 and P6 can be seen as a form of overfitting to the splits and default h-params respectively. Therefore, when selecting h-params of the attacks (e.g. the value of $\lambda$ for LSA attacks and the learning rate for meta attacks) we choose h-params that minimize the test accuracy on average across splits, preventing overfitting to a single split. Moreover, to remedy P6 we poison $\mathcal{V}_l$ which implicitly impacts the bi-level optimization.
Moreover, while fixing the identified pitfalls leads to a fairer evaluation it also decreases the effectiveness of previous attacks. This provided additional motivation to introduce the new baseline attacks. Under the previous evaluation setting, we observe that Meta attacks significantly outperform other attacks (see A.12). However, in the new fairer setting, Meta attacks don't generalize as effectively as linear attacks. This highlights the need for a careful evaluation.

We updated our paper to clarify the connection between pitfalls. We include this discussion in section A.18. We also now identify each pitfall from P1 to P6 in the updated version to help with readability. All updates are shown in blue for clarity.

---

### Meta-Review · Area_Chair_JBQU · 2023-12-12

**Metareview:**

This paper looks at robustness against adversarially-determined noisy labels for nodes in graph data; specifically, it provides a critical take on the literature for poisoning attacks on GNNs.  Evaluation in adversarial ML writ large is notoriously tricky, especially when it comes to evaluating robustness to poisoning attacks.  Reviewers appreciated this paper's take on holes in the poisoning cross GNN literature, in addition to its proposed steps forward.  The paper backs up this discussion of holes by presenting a couple of new attacks based on them; while I'm less excited about the creation of yet another attack or two, this is good empirical validation of the points raised.  The value of this paper is primarily the discussion of a well-researched and -supported set of holes in the way we go about thinking about poisoning graph data -- valuable indeed.

**Justification For Why Not Higher Score:**

Relatively niche area; some valid concerns from reviewers that went unaddressed.

**Justification For Why Not Lower Score:**

Broad support for the paper; poisoning is of increasing interest to the ML community.

---

### Decision · Program_Chairs · 2024-01-16

Accept (poster)